# A Benchmark Dataset to Distinguish Human-Written and Machine-Generated Scientific Papers †

**Mohamed Hesham Ibrahim Abdalla ‡, Simon Malberg ‡⬦, Daryna Dementieva *⬦, Edoardo Mosca ⬦ and Georg Groh ***

School of Computation, Information and Technology, Technical University of Munich, 80333 Munich, Germany; ge96rac@mytum.de (M.H.I.A.); simon.malberg@tum.de (S.M.); edoardo.mosca@tum.de (E.M.)
* Correspondence: daryna.dementieva@tum.de (D.D.); grohg@in.tum.de (G.G.)
† This paper is a substantially extended and revised version of research published in Mosca E.; Abdalla M.H.I.; Basso P.; Musumeci M.; and Groh G. Distinguishing Fact from Fiction: A Benchmark Dataset for Identifying Machine-Generated Scientific Papers in the LLM Era. In Proceedings of the 3rd Workshop on Trustworthy Natural Language Processing (TrustNLP 2023), pages 190–207, Toronto, Canada. Association for Computational Linguistics.
‡ These authors contributed equally to this work.

**Abstract:** As generative NLP can now produce content nearly indistinguishable from human writing, it is becoming difficult to identify genuine research contributions in academic writing and scientific publications. Moreover, information in machine-generated text can be factually wrong or even entirely fabricated. In this work, we introduce a novel benchmark dataset containing human-written and machine-generated scientific papers from SCIgen, GPT-2, GPT-3, ChatGPT, and Galactica, as well as papers co-created by humans and ChatGPT. We also experiment with several types of classifiers—linguistic-based and transformer-based—for detecting the authorship of scientific text. A strong focus is put on generalization capabilities and explainability to highlight the strengths and weaknesses of these detectors. Our work makes an important step towards creating more robust methods for distinguishing between human-written and machine-generated scientific papers, ultimately ensuring the integrity of scientific literature.

**Keywords:** text generation; large language models; machine-generated text detection





## 1. Introduction

Generative *Natural Language Processing* (NLP) systems—often based on *Large Language Models* (LLMs) [1–3]—have experienced significant advancements in recent years, with state-of-the-art algorithms generating content that is almost indistinguishable from human-written text [1,4–7]. This progress has led to numerous applications in various fields, such as chatbots [8], automated content generation [9], and even summarization tools [10]. However, these advancements also raise concerns regarding the integrity and authenticity of academic writing and scientific publications [11,12].

It is indeed increasingly difficult to differentiate genuine research contributions from artificially generated content. Moreover, we are at an increased risk of including factually incorrect or entirely fabricated information [13,14]. Reliably identifying machine-generated scientific publications becomes, thus, crucial to maintaining the credibility of scientific literature and fostering trust among researchers.

This work introduces a novel benchmark to address this issue. Our contribution—also briefly sketched in Figure 1—can be summarized as follows:

(1) We present a dataset comprising human-written and machine-generated scientific documents from various sources: SCIgen [15], GPT-2 [4], GPT-3 [1], ChatGPT [8], and Galactica [16]. We also include a set of human–machine co-created documents resembling scientific documents with both human-written and machine-paraphrased

texts. Each document includes an *abstract*, *introduction*, and *conclusion* in a machine-readable format. While real titles were used to generate articles for the dataset, there is no title intersection between real and machine-generated papers in our dataset.

(2) We experiment with several classifiers—bag-of-words-based classifiers, RoBERTa [17], Galactica [16], GPT-3 [1], DetectGPT [18], ChatGPT [8], and a proposed novel classifier learning features using an LLM and Random Forest [19]—assessing their performance in differentiating between human-written and machine-generated content. We also assess each classifier's generalization capabilities on out-of-domain data and human–machine co-created papers to obtain a more accurate estimate of the likely real-world performance of the different classifiers.

(3) We explore explainability insights from different classifiers ranging from word-level explanations to more abstract concepts to identify typical differences between human-written and machine-generated scientific papers.

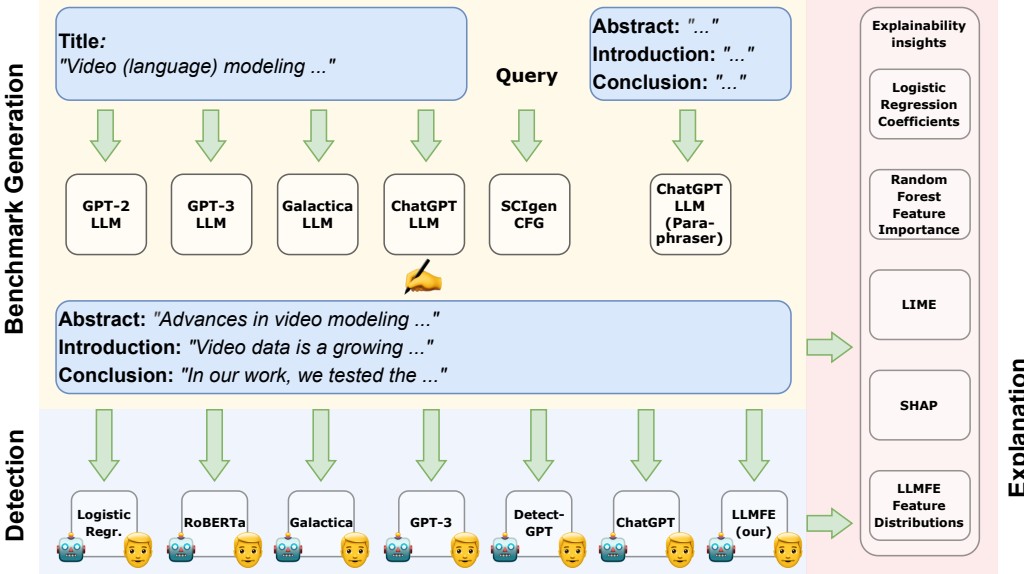

**Figure 1.** This work's overview. Six methods are used to machine-generate papers, which are then mixed with human-written ones to create our benchmark dataset. Seven models are then tested as baselines to identify the authorship of a given output.

We release our benchmark dataset, baseline models, and testing code to the public to promote further research and aid the development of more robust detection methods. (https://huggingface.co/datasets/tum-nlp/IDMGSP) (accessed on 31 July 2023). This work extends a previously published study [20].

## 2. Related Work

### 2.1. Machine-Generated Text Detection Benchmarks

Since the significant improvement of text generation models, the potential danger and harm of machine-generated text has been acknowledged by NLP researchers. For this reason, existing generations of generative models have been tested to create texts in various domains to compile human-written vs. machine-generated benchmarks.

One of the first datasets and models to detect neural generated texts in a news domain was Grover [7]. The Grover model for neural news generation was based on GPT-2 [4] and was used to create a benchmark for neural news detection. In addition, for the news domain, a dataset for automatic detection of machine-generated news headlines was created [21]. The machine-generated headlines were also created with GPT-2. Beyond fake news, the detection of generated scientific articles received attention as well, leading to the first task benchmarks introduced in [22].

With the increasing capabilities of LLMs, new benchmarks appeared recently, covering several neural text generators and domains. In [23], the M4 (multi-generator, multi-domain, and multi-lingual) dataset was presented. It covers various kinds of topics—Wikipedia articles, question-answering posts, news, and social posts—in six languages. In the MGTBench benchmark [24], LLMs were evaluated on several different question-answering datasets. Finally, the DeepfakeTextDetect dataset [25] covers news article writing, story generation, scientific writing, argument generation, and question-answering.

### 2.2. Scientific Publication Corpora: Human and Machine-Generated

The ACL Anthology (https://aclanthology.org) (accessed on 24 April 2023). Ref. [26] and arXiv [27] are widely used resources for accessing scientific texts and their associated metadata. However, these databases do not provide structured text for scientific documents, necessitating the use of PDF parsers and other tools to extract text and resolve references. Several efforts have been made to develop structured text databases for scientific documents [28–30].

Despite progress in generating text, machine-generated datasets for scientific literature remain limited. A recent study by Kashnitsky et al. [31] compiled a dataset including shortened, summarized, and paraphrased paper abstracts and excerpts, as well as text generated by GPT-3 [1] and GPT-Neo [32]. The dataset lists retracted papers as machine-generated, which may not always be accurate, and only includes excerpts or abstracts of the papers.

Liyanage et al. [22] proposed an alternative approach, in which they generated papers using GPT-2 [4] and arXiv-NLP (https://huggingface.co/lysandre/arxiv-nlp) (accessed on 24 April 2023). However, their dataset was limited to only 200 samples, which were restricted to the fields of Artificial Intelligence and Computation and Language.

### 2.3. Generative NLP for Scientific Articles

Generative NLP for scientific publications has evolved significantly in recent years. Early methods, such as SCIgen [15], used *Context-Free-Grammars* (CFG) to fabricate computer science publications. These often contain nonsensical outputs due to CFG's limited capacity for generating coherent text.

With the advent of attention, transformers [33] and LLMs [1] have paved the way for more sophisticated models capable of generating higher-quality scientific content. Some known (both opensourced and closed) LLMs—such as GPT-3 [1], ChatGPT [8], Bloom [2], LLaMa-2 [6], and PaLM-2 [34]—are built for general purposes. Others, instead, are domain-specific and specialized for generating scientific literature. Popular examples in this category are SciBERT [35] and Galactica [16].

Both general and domain-specific models have shown outstanding results in various scientific tasks, demonstrating their potential to generate coherent and contextually relevant scientific text. Consequentially, the same technology has been applied to other domains, including writing news articles [7], producing learning material [36], and creative writing [37]. Moreover, in education, the usage of advanced LLMs showed already promising results in providing "live" help in the teaching process [38]. For such use cases, it is important to develop trustworthy machine-generation technologies, able to provide both factually correct information as well as display fluency in communication with the users.

### 2.4. Detection of Machine-Generated Text

The ability to automatically generate convincing content has motivated researchers to work on its automatic detection, especially given its potential implications for various domains.

Several approaches to detecting machine-generated text have emerged, employing various techniques. In [39], a survey of the methods for machine-generated text detection was presented. One solution is utilizing hand-crafted features [40]. In addition, linguistic-based and bag-of-words features can be quite powerful and well-explainable baselines [41].

The topology of attention masks was proven to be one of the efficient methods to detect neural-generated texts in [42]. Finally, neural features in combination with supervised models can be trained to distinguish between human and machine-generated content [41,43,44].

Alternative approaches explore using the probability distribution of the generative model itself [18] or watermarking machine-generated text to facilitate detection [45].

### 2.5. Detection of Machine-Generated Scientific Publications

As we have seen in Section 2.4, several general-purpose solutions exist aiming to detect NLP-generated text. The detection of automatically generated scientific publications, however, is an emerging subarea of research with a large potential for improvement.

Previous approaches have primarily focused on identifying text generated by SCIgen [15] using hand-crafted features [46,47], nearest neighbor classifiers [48], and grammar-based detectors [49]. More recent studies have shown promising results in detecting LLM-generated papers using SciBERT [50], DistilBERT [51], and other transformer-based models [22,52]. Nonetheless, these approaches have mostly been tested only on abstracts or a substantially limited set of paper domains.

With the appearance of ChatGPT [8], several studies were dedicated to evaluating how good this model can be in generating scientific papers. In [53], it was shown that human annotators are incapable of identifying ChatGPT-generated papers. Since ChatGPT can not only be used to generate papers from scratch but also to paraphrase them, a method to identify the polish-ratio of ChatGPT in a piece of text was proposed in [54].

In the end, we can see the necessity for an explainable and robust detector able to detect machine-generated and edited articles from the most recent LLMs. With this work, we are aiming to make a step towards the creation of such automated detectors.

## 3. Benchmark Dataset

In this section, we delve into the construction of our benchmark dataset, which comprises human-written, machine-generated, and human–machine co-created scientific papers. Often, for simplicity, we refer to these groups as *real*, *fake*, and *co-created*, respectively. In Section 3.1, we elaborate on the process we followed to extract data from the PDF documents of real papers. In Section 3.2, we describe our prompting pipelines and how we utilized various generators to produce fake scientific papers. In Section 3.3, we explain our approach to generating human–machine co-created papers.

Table 1 offers an overview of our dataset, including sources and numbers of samples and tokens.

**Table 1.** Data sources included in our dataset and their respective sizes.

| Source | Quantity | Tokens |
|---|---|---|
| arXiv parsing 1 (real) | 12 k | 13.4 M |
| arXiv parsing 2 (real) | 4 k | 3.2 M |
| SCIgen (fake) | 3 k | 1.8 M |
| GPT-2 (fake) | 3 k | 2.9 M |
| Galactica (fake) | 3 k | 2.0 M |
| ChatGPT (fake) | 3 k | 1.2 M |
| GPT-3 (fake) | 1 k | 0.5 M |
| ChatGPT (paraphrased real) | 4 k | 3.5 M |
| Total real (extraction) | 16 k | 16.6 M |
| Total fake (generators) | 13 k | 8.4 M |
| Total co-created (paraphrased) | 4 k | 3.5 M |
| Total | 33 k | 28.5 M |

### 3.1. Real Papers Collection

To collect human-written—or *real*—scientific papers for our dataset, we source them from the arXiv dataset [27] hosted on Kaggle (https://www.kaggle.com/datasets/Cornell-

University/arxiv (accessed on 24 April 2023)). We exclude scientific papers published after ChatGPT (after November 2022) to avoid machine-generated papers leaking into our *real* dataset. While it is still possible that some of the remaining papers were machine-generated, we deem this to be highly unlikely and only affect a negligibly small number of papers, if at all, given the lower accessibility and quality of generators before ChatGPT.

The arXiv dataset provides comprehensive metadata, including title, abstract, publication date, and category. However, the introduction and conclusion sections are not part of the metadata, which implies the need for PDF parsing to extract these sections. From the metadata, each paper's ID and version are utilized to construct the document path and retrieve the corresponding PDF from the publicly accessible Google Cloud Storage bucket. Each PDF is then fed to the PyMuPDF [55] library to be parsed and to extract the relevant content. Unfortunately, parsing PDFs is known to be very challenging. This is particularly true for a double-column format, which many scientific papers have. Despite having tested several heuristic rules to identify and extrapolate the correct sections, the process can still fail at times. We discard data points where the parsing was unsuccessful.

The resulting set includes 12,000 real papers. Furthermore, we collect an additional 4000 samples undergoing a different parsing procedure. The intention is to ensure there are no recognizable parsing artifacts that inadvertently ease the detection process (see Section 4).

### 3.2. Fake Papers Generation

For the *fake* component of our dataset, we employ several models to generate abstracts, introductions, and conclusions based on scientific paper titles. The overview of the models used for generation is illustrated in Figure 2. The titles of the real papers sourced from the arXiv database (see Section 3.1) serve as prompts for the models to generate the target sections—i.e., ***abstract***, ***introduction***, and ***conclusion***.

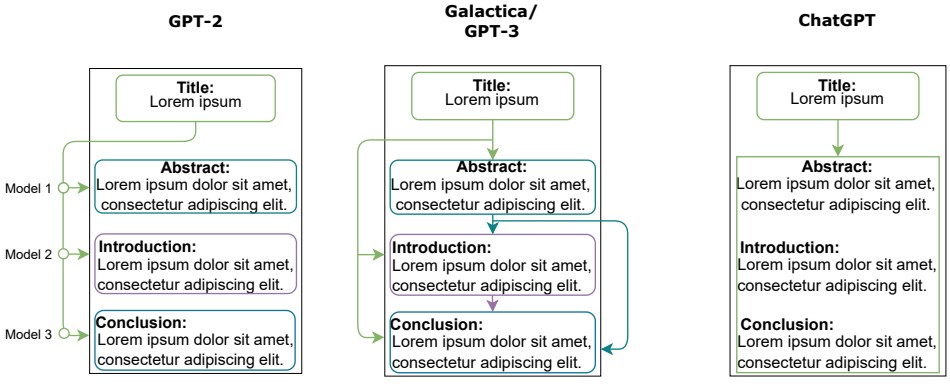

(**a**) GPT-2 generation          (**b**) Galactica/GPT-3 generation          (**c**) ChatGPT generation

**Figure 2.** Generation pipeline used for each model. For GPT-2 (**a**), the abstract, introduction, and conclusion sections are generated by three separately fine-tuned model instances, each based solely on the paper title. In the case of Galactica and GPT-3 (**b**), each section is generated conditioning on the previous sections. Finally, ChatGPT's generation sequence (**c**) requires only the title to generate all the necessary sections at once.

To create fake scientific papers, we fine-tune GPT-2 and GPT-3 instances [1,4] and also leverage SCIgen [15], Galactica [16], and ChatGPT [8]. For each model—as shown in Figure 2—we employ a unique prompting/querying strategy to produce the desired paper sections.

This combination of models, ranging from CFG to state-of-the-art LLMs, aims to generate a diverse set of artificially generated scientific papers. Concrete examples of generated papers can be found in Appendix A.

### 3.2.1. SCIgen

Alongside the papers produced by the various LLMs, our fake dataset incorporates documents generated by SCIgen [15]. Although the quality of CFG-generated text is rather low and hence straightforward to identify, it remains relevant to ensure that current detectors can distinguish machine-generated papers even if poorly written and containing nonsensical content. Stribling and Aguayo [56] show that such papers have been accepted in scientific venues in the past.

Prompting SCIgen is done simply by running it as an offline script (https://github.com/soerface/scigen-docker) (accessed on 24 April 2023) which generates all the needed sections, including the title. The entire paper in LaTeXformat is generated as a result.

### 3.2.2. GPT-2

We fine-tune three distinct GPT-2 base (https://huggingface.co/gpt2) (accessed on 24 April 2023) models (124 M parameters) [4] to individually generate each section based on the given title. The models are trained in a seq2seq fashion [57], with the training procedure spanning six epochs and incorporating 3500 real papers. When encountering lengthy inputs, we truncate those exceeding 1024 tokens, potentially resulting in less coherent introductions and conclusions. Abstracts remain more coherent as they typically fall below this threshold. We release these separately fine-tuned GPT-2 instances to generate abstract (https://huggingface.co/tum-nlp/IDMGSP-GPT-2-ABSTRACT) (accessed on 31 July 2023), introduction (https://huggingface.co/tum-nlp/IDMGSP-GPT-2-INTRODUCTION) (accessed on 31 July 2023), and conclusion (https://huggingface.co/tum-nlp/IDMGSP-GPT-2-CONCLUSION) (accessed on 31 July 2023) for public usage and investigation.

**Hyperparameters:** For training, we use a batch size of 16 across all six epochs. We set the `max_new_token` to 512, `top_k` to 50, and `top_p` to 0.5 for all three models.

**Post-processing:** We remove generated "$\backslash n$" characters and any extra sections not explicitly mentioned in the prompt. Additionally, we remove incomplete sentences preceding the start of a new sentence. These are indeed common artifacts of GPT-2 and are easily identifiable by lowercase letters.

Although our GPT-2 model is specifically fine-tuned for the task, generating long pieces of text occasionally results in less meaningful content. Moreover, we observe that decoupling the generation of sections can lead to inconsistencies among the generated sections within the papers.

### 3.2.3. Galactica

Galactica is trained on a large corpus of scientific documents [16]. Therefore, it is already well-suited for the task of generating scientific papers. To facilitate the generation of coherent long-form text, we divide the generation process into smaller segments, with each section relying on preceding sections for context. For instance, while generating a conclusion, we provide the model with the title, abstract, and introduction as concatenated text.

**Hyperparameters:** We use Galactica base (https:// huggingface.co/facebook/galactica-1.3b) (accessed on 24 April 2023) (1.3 B parameters) [16] to generate each paper section based on the previous sections. The complete set of hyperparameters can be found in Table A1 in the Appendix A. Additionally, we enforce max length left padding. Due to the limited model capacity, restriction of the output number of tokens is necessary to avoid the hallucination risk introduced by long text generation.

**Post-processing:** To ensure completeness and coherence in the generated text, we devise a generation loop that meticulously assesses the quality of the output. For example, if the generated text lacks an `<EOS>` (end-of-sentence) token, the model is prompted to regenerate the text. Furthermore, we eliminate any special tokens introduced by Galactica during the process.

While Galactica base has 1.3 B parameters, it is still smaller than ChatGPT, which can result in less coherent outputs when generating longer text segments. As a result,

prompting the model to generate a specific section with preceding sections as context yields better outcomes compared to providing only the title as context and requesting the model to generate all three sections simultaneously.

### 3.2.4. ChatGPT

To generate a cohesive document, we prompt ChatGPT (https://help.openai.com/en/articles/6825453-chatgpt-release-notes, release from 15 December 2022) [8] with "*Write a document with the title* [TITLE], *including an abstract, an introduction, and a conclusion*", substituting [TITLE] with the desired title utterance. ChatGPT's large size (20B parameters) and strong ability to consider context eliminate the necessity of feeding previous output sections into the prompt for generating newer ones.

**Hyperparameters:** For the entire generation process, we use the default temperature of 0.7.

Despite not being explicitly trained for scientific text generation, ChatGPT can produce extensive, human-like text in this domain. This capability likely stems from the model's large size, the extensive datasets it was trained on, and the incorporation of reinforcement learning with human feedback.

### 3.2.5. GPT-3

We fine-tune an instance of GPT-3 (`text-curie-001`, 6.7 B parameters) [1] with 178 real samples. Output papers generated through an iterative cascade process (as with Galactica) present a much higher quality than those forged in a single step (as with ChatGPT). Hence, we opt for the former strategy for GPT-3.

**Pre/Post-Processing:** To force the generation of cleaner outputs, we add an `<END>` token at the end of each input used for fine-tuning. GPT-3 mimics this behavior and predicts this token as well, so we remove every token added after generation `<END>`.

While still not on par with ChatGPT-generated outputs, we report a high quality for GPT-3-crafted papers.

### 3.3. Co-Created Papers Generation

The co-created component of our dataset mimics papers written by humans and models concurrently, a combination that is likely to appear in practice. That means texts originally written by either a human or an LLM and subsequently extended, paraphrased, or otherwise adjusted by the other. To create such papers at scale, we take a set of 4000 real papers from our TEST dataset (see Table 2) and paraphrase them with ChatGPT [8]. To stay within ChatGPT's context length limits, we paraphrase each paper section—i.e., *abstract*, *introduction*, and *conclusion*—in a separate prompt. We then construct co-created papers with varying shares of human and machine input by combining original and paraphrased sections as shown in Figure 3.

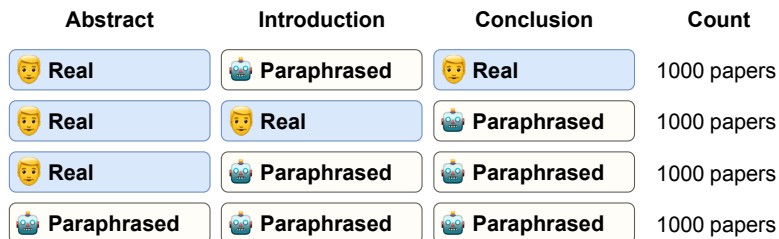

**Figure 3.** Our co-created test dataset TEST-CC contains 4000 papers with varying shares of *real* and ChatGPT-paraphrased sections.

**Table 2.** Overview of the datasets used to train and evaluate the classifiers. Each column represents the number of papers used per source. Concerning *real* papers, unless indicated, we use samples extracted with parsing 1 (see Section 3.1).

| Dataset | arXiv (Real) | ChatGPT (Fake) | GPT-2 (Fake) | SCIgen (Fake) | Galactica (Fake) | GPT-3 (Fake) | ChatGPT (Co-Created) |
|---|---|---|---|---|---|---|---|
| Standard train (TRAIN) | 8 k | 2 k | 2 k | 2 k | 2 k | - | - |
| Standard train subset (TRAIN-SUB) | 4 k | 1 k | 1 k | 1 k | 1 k | - | - |
| TRAIN without ChatGPT (TRAIN-CG) | 8 k | - | 2 k | 2 k | 2 k | - | - |
| TRAIN plus GPT-3 (TRAIN + GPT3) | 8 k | 2 k | 2 k | 2 k | 2 k | 1.2 k | - |
| Standard test (TEST) | 4 k | 1 k | 1 k | 1 k | 1 k | - | - |
| Out-of-domain GPT-3 only (OOD-GPT3) | - | - | - | - | - | 1 k | - |
| Out-of-domain real (OOD-REAL) | 4 k (parsing 2) | - | - | - | - | - | - |
| ChatGPT only (TECG) | - | 1 k | - | - | - | - | - |
| Co-created test (TEST-CC) | - | - | - | - | - | - | 4 k |

**Hyperparameters:** For paraphrasing, we use OpenAI's `gpt-3.5-turbo-0613` model and set the temperature to 1.0 to achieve the largest deviation from the original human-written text.

## 4. Detection Experiments

In this section, we conduct experiments about identifying the source of a given paper—i.e., determining whether it is *fake* or *real*. We further investigate the ability of our baseline classifiers to detect co-created papers with varying degrees of *fake*—i.e., paraphrased—content. We start by defining data splits and subsets for training and testing, which are useful to evaluate generalization capabilities. Next, we outline the classifiers used as baselines to measure performance on the benchmark task. Finally, we examine the detection performance of the classifiers, investigate the obtained explanations, and apply additional post hoc explainability methods to the classifiers to gain deeper insights into the detection process.

### 4.1. Data Splits and Generalization Tests

We divide our dataset (displayed in Table 1) into *standard train* and *standard test* sets for training and testing our classifiers, respectively. Furthermore, we aim to evaluate models on out-of-domain test data. To achieve this, we create various data subsets by applying different splits to our benchmark. All the splits utilized for our experiments are detailed in Table 2. For instance, the reader can observe the composition of a data split with no access to ChatGPT samples (TRAIN-CG) and test sets composed only of differently-parsed real papers (OOD-REAL), only ChatGPT papers (OOD-CG), or only GPT-3 ones (OOD-GPT3).

### 4.2. Classifiers

We build and evaluate seven classifiers to perform the downstream task of classifying scientific papers as *fake* or *real* based on their content (abstract, introduction, and conclusion sections)—we remind the reader that all paper titles are *real* and will therefore not serve as input to the classifiers. To obtain an understanding of the difficulty of this classification task, we train two simple bag-of-words-based classifiers, Logistic Regression (LR) [58] and Random Forest (RF) [19]. Further, we fine-tune GPT-3 [1], Galactica [16], and RoBERTa [17] for this detection task. Lastly, we use a ChatGPT-based classifier without fine-tuning and a novel classifier that we call Large Language Model Feature Extractor (LLMFE) that learns explainability features using an LLM and then performs classification with Random Forest.

To accommodate memory and API limitations, we impose a restriction on the input tokens for GPT-3, Galactica, and RoBERTa by truncating texts after a certain number of tokens (details described in the following sections per model). However, since the average length of the combined input sections is about 900 tokens, which is less than the truncation limit, this constraint does not lead to significant information loss.

### 4.2.1. Bag-of-Words Classifier

As the simplest classifiers, we evaluate Random Forest [19] and Logistic Regression [58] on TF-IDF [59] features. This is to obtain an indication of the difficulty of the classification task—i.e., whether there is any classification signal in word frequencies alone or the detection of *fake* scientific papers requires more complex features. With Random Forest and Logistic Regression, we can explain the results by examining feature importance and learned coefficients.

**Hyperparameters:** We use the Random Forest and Logistic Regression implementations in scikit-learn [60] with default hyperparameters. We create features based on n-grams. A comparison of accuracies when using 1-grams, 2-grams, or a combination of both can be found in Table A2 in the appendix. In the following, we will report results based on 1-grams as these yielded the highest accuracy scores.

### 4.2.2. GPT-3

We fine-tune a GPT-3 [1] Ada model (`text-ada-001`, 350 M parameters) for the classification task. GPT-3 is fine-tuned in a causal manner, where the model is prompted with the concatenated paper sections along with their corresponding label. This is set up as a binary classification where the output is a single token indicating whether the paper is *real* (0) or *fake* (1). During inference, the model generates a single token based on the sections of a given paper.

As fine-tuning GPT-3 models requires a paid API, we train it only on a smaller subset of our dataset (TRAIN-SUB) shown in Table 2. We limit the number of input tokens to 2048 while retaining the default hyperparameters provided by the API.

### 4.2.3. Galactica

We adapt Galactica-mini (https://huggingface.co/facebook/galactica-125m) (accessed on 24 April 2023) [16] from a causal language model that predicts probabilities for each word in the vocabulary to a binary classifier with an output layer that predicts probabilities for two labels: *fake* and *real*.

The model is provided with all sections concatenated together with the corresponding label. Galactica, being a causal language model, generates a probability distribution spanning the entire vocabulary in its output. Nevertheless, this approach incurs significant memory usage, particularly when employed as a classifier. Therefore, we opted to retrain the output layer to yield a probability distribution for binary outcomes.

**Hyperparameters:** To cope with memory constraints, we limit the number of input tokens to 2048. Additionally, we adjust the batch size to 2 with gradient accumulation steps of 4 and enabled mixed precision. Furthermore, we set the number of epochs to 4, weight decay to 0.01, and warm-up steps to 1000. Our initial learning rate is $5 \times 10^{-6}$.

### 4.2.4. RoBERTa

We fine-tune RoBERTa base (125 M parameters) (https://huggingface.co/roberta-base) (accessed on 24 April 2023) [17] for the classification task. RoBERTa is limited to 512 input tokens, meaning that all text exceeding this limit is ignored. Our dataset exceeds this constraint for many entries. We choose to address the problem by fine-tuning three separate RoBERTa models to classify the three sections individually rather than retraining the input layer by enlarging the input size. https://huggingface.co/tum-nlp/IDMGSP-RoBERTa-TRAIN-ABSTRACT (accessed on 31 July 2023) (https://huggingface.co/tum-nlp/IDMGSP-RoBERTa-TRAIN-INTRODUCTION) (accessed on 31 July 2023) (https://huggingface.co/tum-nlp/IDMGSP-RoBERTa-TRAIN-CONCLUSION) (accessed on 31 July 2023) We take the majority vote from three model instances as the final output for each sample. We prompt each model with the capitalized name of the section plus the content of the latter, e.g., "*Abstract: In this paper . . .*".

**Hyperparameters:** To fine-tune the RoBERTa base, we set the number of epochs to 2, weight decay to 0.001, and batch size to 16. As with Galactica, the initial learning rate is $5 \times 10^{-6}$, and the warmup steps 1000.

### 4.2.5. DetectGPT

We evaluate DetectGPT [18] as another classifier as it has been shown to detect LLM-generated texts with high accuracy.

**Hyperparameters:** We use DetectGPT's default configuration and code (https://github.com/BurhanUlTayyab/DetectGPT) (accessed on 15 May 2023).

### 4.2.6. ChatGPT

To obtain natural-language explanations for classification directly, we prompt Chat-GPT [8] via the OpenAI API. With this, we determine whether a scientific paper is *fake* or *real* and retrieve an explanation for its decision. The prompts include the concatenated sections, each beginning with the section name (e.g., "*Abstract:\nIn this paper . . .*"), and task instructions. We compare the detection performance of four different prompting styles:

(1) **Input-Output Prompting (IO):** First, return the prediction (i.e., *fake* or *real*). Second, follow up with an explanation of the reasons for the prediction.
(2) **Chain-of-Thought Prompting (CoT) [61]:** First, return a sequence of thoughts on whether the paper is more likely *fake* or *real*. Second, return the final prediction.
(3) **Indicator Prompting (IP):** First, return a set of observations indicating that the paper was written by a human. Second, return a set of observations indicating that the paper was generated by a machine. Third, return the final prediction.
(4) **Few-Shot Prompting (FS) [1]:** Perform Input-Output Prompting but include a set of 6 annotated examples—one example from each generator and one *real* example—in the prompt (i.e., scientific papers with their abstract, introduction, conclusion, and *fake* or *real* label).

On our specific task, we observe the best classification results for the IO prompting style. Hence, we will only report accuracy scores for this prompting style in the following. For a detailed accuracy comparison of the different prompting styles, see Table A3 in the appendix. When using CoT prompting, there is a large number of instances where ChatGPT refuses to return a definite class label (*real* or *fake*) but instead returns *unknown*. We treat these results as incorrect answers and thus observe low accuracy scores for CoT prompting. We did not observe this behavior for the other prompting styles.

**Hyperparameters:** For classification, we use OpenAI's `gpt-3.5-turbo-0613` model with the default temperature of 0.7. Only for Few-Shot Prompting, we prompt the `gpt-3.5-turbo-16k-0613` model as a larger context length is needed. We do not perform task-specific fine-tuning. Due to API limitations, we classify only 100 randomly sampled papers from each test set using each of the four prompting styles. During implementation, we also experimented with larger samples and observed consistent classification accuracy scores independent of the sample size.

### 4.2.7. Large Language Model Feature Extractor (LLMFE)

Finally, we introduce and evaluate a novel explainable classifier LLMFE that learns human-understandable features using an LLM and an approach inspired by contrastive learning [62]. These features can range from very low-level (e.g., occurrences of a specific word) to very high-level (e.g., logical conclusiveness of argumentation). Figure 4 shows how LLMFE works conceptually. Training this classifier follows a four-step process:

(1) **Feature Engineering:** The LLM is presented with a pair of one *real* and one *fake* scientific paper and instructed to describe a list of features that would best distinguish these papers from each other. As we score each feature on a range of 0 to 10, we further instruct the LLM to label the meaning of the extreme ends of this scale for each feature

to avoid ambiguity. This prompt is repeated for `n_pairs` times to extract multiple different sets of features based on different example pairs.

(2) **Feature Consolidation:** As the previous step may have generated a large number of features, many of which are duplicates or semantically similar, we consolidate the extracted features into a smaller feature set. This is done by vectorizing each feature description using embeddings and performing hierarchical/agglomerative clustering [63] on the embeddings. We then manually investigate the cluster dendrogram and define a distance threshold `d_thres`. We finally merge all features less than `d_thres` apart from each other and represent each cluster through the feature closest to the cluster centroid. If `d_thres` is chosen carefully, this results in a significantly smaller, semantically diverse, and duplicate-free feature set. More detailed illustrations of this step can be found in Appendix B.4.

(3) **Feature Scoring:** The LLM is presented with an abstract, introduction, and conclusion of a scientific paper and descriptions of all features in the feature set. It is then instructed to assign an integer value from 0 to 10 to each feature that most accurately describes the scientific paper. This prompt is repeated for each example in the training dataset of size `n_sample`.

(4) **Classifier Training:** The previous steps resulted in a structured dataset of `n_sample` examples with one integer value for each feature in the learned feature set. Further, class labels (i.e., *real* or *fake*) are known. This dataset is used to train a Random Forest [19] classifier that learns to detect papers based on the features described by the LLM.

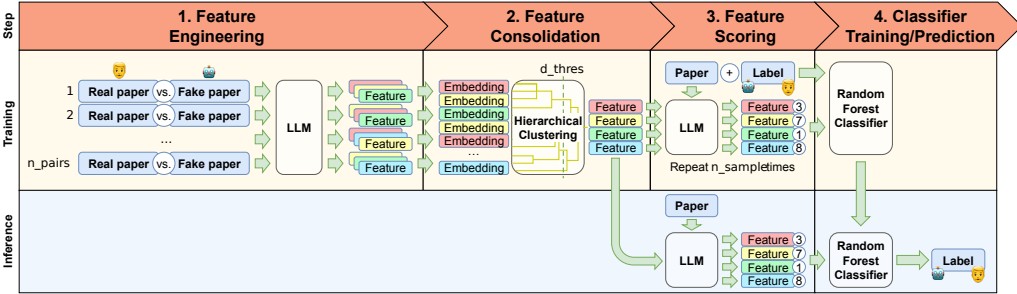

**Figure 4.** LLMFE follows a four-step process: (1) Generate features suitable for distinguishing *real* and *fake* papers using the LLM based on multiple pairs of one *real* and one *fake* paper each. (2) Remove duplicate features through hierarchical clustering on embeddings of the feature descriptions. (3) Score scientific papers along the remaining features using the LLM. (4) Finally, train a Random Forest Classifier to predict the *real* or *fake* label based on the feature scores.

Throughout the first three steps, the LLM is made aware of its overall goal of distinguishing *real* and *fake* scientific papers through the prompt instructions. We add this context information to best exploit the LLM's general world understanding obtained through extensive pre-training and to compensate for the relatively small sample sizes used for training. Inference on the test dataset then requires only two steps:

**(1)** **Feature Scoring:** Similar to the Feature Scoring step during training, a set of new papers is scored along the learned features.

**(2)** **Classifier Prediction:** The class label of the new papers is predicted using the trained Random Forest classifier.

**Hyperparameters:** Our LLMFE implementation uses OpenAI's `gpt-3.5-turbo-0613` with the default temperature of 0.7 for the Feature Engineering step and `gpt-3.5-turbo-16k -0613` with a temperature of 0.0—for deterministic behavior—for the Feature Scoring step. We set `n_pairs=100` and obtained 884 features from the Feature Engineering step. For the Feature Consolidation step, we create embeddings of the feature descriptions with OpenAI's `text-embedding-ada-002` and `chunk_size=1000`. We apply agglomerative clustering from Scipy's [64] linkage implementation with a cosine distance metric and calculate

the `average` distance between clusters. We chose `d_thres=0.05` as this resulted in a convenient balance between de-duplication and semantic feature diversity, yielding a final set of 83 features. We finally trained a Random Forest classifier with scikit-learn's [60] default hyperparameters on 600 papers from the TRAIN dataset (300 *real* papers and 60 *fake* papers from each generator).

*4.3. Performance*

Table 3 presents a summary of the accuracy scores achieved by our models on various splits. Given the significance of evaluating generalization to unseen generators, we highlight out-of-domain settings in blue. We exclude experiments entailing training GPT-3 on TRAIN + GPT3 and TRAIN-CG due to limited OpenAI API credits. Results of our fine-tuned models and LLMFE are also compared with DetectGPT as an existing zero-shot detection baseline [18], ChatGPT, and our Logistic Regression (LR) and Random Forest (RF) classifiers trained on 1-gram TF-IDF features.

**Table 3.** Experiment results reported with accuracy metric. Out-of-domain experiments, i.e., evaluation on unseen generators, are highlighted in blue. Highest values per test set are highlighted in bold. (*) ChatGPT-IO and LLMFE accuracies have been evaluated on randomly sampled subsets of 100 scientific papers per test set due to API limits.

| Model | Train Dataset | TEST | OOD-GPT3 | OOD-REAL | TECG | TEST-CC |
|---|---|---|---|---|---|---|
| LR-1gram (tf-idf) (our) | TRAIN | 95.3% | 4.0% | 94.6% | 96.1% | 7.8% |
| LR-1gram (tf-idf) (our) | TRAIN + GPT3 | 94.6% | 86.5% | 86.2% | 97.8% | 13.7% |
| LR-1gram (tf-idf) (our) | TRAIN-CG | 86.6% | 0.8% | 97.8% | 32.6% | 1.2% |
| RF-1gram (tf-idf) (our) | TRAIN | 94.8% | 24.7% | 87.3% | **100.0%** | 8.1% |
| RF-1gram (tf-idf) (our) | TRAIN + GPT3 | 91.7% | 95.0% | 69.3% | **100.0%** | 15.1% |
| RF-1gram (tf-idf) (our) | TRAIN-CG | 97.6% | 7.0% | 95.0% | 57.0% | 1.7% |
| Galactica (our) | TRAIN | 98.4% | 25.9% | 95.5% | 84.0% | 6.8% |
| Galactica (our) | TRAIN + GPT3 | 98.5% | 71.2% | 95.1% | 84.0% | 12.0% |
| Galactica (our) | TRAIN-CG | 96.4% | 12.4% | 97.6% | 61.3% | 2.4% |
| RoBERTa (our) | TRAIN | 72.3% | 55.5% | 50.0% | **100.0%** | 63.5% |
| RoBERTa (our) | TRAIN + GPT3 | 65.7% | **100.0%** | 29.1% | **100.0%** | **75.0%** |
| RoBERTa (our) | TRAIN-CG | 86.0% | 2.0% | 92.5% | 76.5% | 9.2% |
| GPT-3 (our) | TRAIN-SUB | **100.0%** | 25.9% | 99.0% | **100.0%** | N/A |
| DetectGPT | - | 61.5% | 0.0% | **99.9%** | 68.7% | N/A |
| ChatGPT-IO (our) (*) | - | 69.0% | 49.0% | 89.0% | 0.0% | 3.0% |
| LLMFE (our) (*) | TRAIN + GPT3 | 80.0% | 62.0% | 70.0% | 90.0% | 33.0% |

Our simplest models, LR and RF, already achieve accuracy scores greater than 90% on the TEST dataset, suggesting that the classification task of distinguishing *real* and *fake* scientific papers is rather easy to learn if trained on comparable scientific papers. However, evaluated against out-of-domain scientific papers, accuracy scores drop significantly. All models perform poorly on out-of-domain papers generated by GPT-3 curie (OOD-GPT3). This result supports the findings of previous studies by Bakhtin et al. [43], which indicate that models trained on specific generators tend to overfit and perform poorly on data outside their training distribution. However, after training our Galactica and RoBERTa models with GPT-3 examples (TRAIN + GPT3), the models achieve higher accuracies (71% and 100%, respectively). A similar behavior can be observed for the LR and RF classifiers.

All models, except RoBERTa, perform poorly when detecting human–machine co-created papers (TEST-CC). Seeing papers generated by ChatGPT and GPT-3 during training each noticeably improves the detection accuracy for all models, presumably because these examples are most similar to the ChatGPT-paraphrased papers that are part of the TEST-CC dataset. RoBERTa still achieves an accuracy of 75%, which is remarkable given that many examples only contain a relatively low share of machine-generated text. This seems to be due to a high-recall bias of the trained RoBERTa model, which achieves comparatively high accuracy scores on datasets that only contain *fake* papers (i.e., OOD-GPT3, TECG) but lower scores on the remaining datasets that also contain *real* papers. GPT-3 and DetectGPT have not been evaluated against TEST-CC due to limited computing resources and API credits.

Models that were not fine-tuned to the classification task, DetectGPT and ChatGPT, perform noticeably worse than the fine-tuned models. Our ChatGPT-based LLMFE outperforms ChatGPT on all test datasets except OOD-REAL, indicating that LLM's detection abilities can be enhanced with a systematic prompting approach and guidance. In particular, we observe great improvements in the more sophisticated texts in our TECG and TEST-CC datasets. This may be because of the more high-level features identified by LLMFE—e.g., those that capture a paper's overall coherence.

It is worth noting that our RoBERTa model exhibits excellent results when evaluated on a dataset of ChatGPT-generated papers (TECG). The model achieves an accuracy of 77% without prior training on a similar dataset (TRAIN-CG), and 100% accuracy when a similar dataset is included in the training (TRAIN). These results outperform Galactica in both scenarios.

The overall good results on OOD-REAL—i.e., real paper processed with a different parser—indicate that our models are not exploiting any spurious artifact introduced during the parsing procedure. DetectGPT notably overfits papers generated with GPT-2 and deems most samples coming from a different source as real. Indeed, it performs well on OOD-REAL (100%) and poorly on OOD-GPT3 (0%).

### 4.4. Explainability Insights

The different types of classifier models provide a rich set of explainability insights that help us understand what characterizes *real* and *fake* scientific papers, respectively. LR and RF classifiers trained on TF-IDF 1-grams provide insights into individual words. For Galactica, RoBERTa, and GPT-3, we extract insights on more complex features of word combinations. Lastly, LLMFE learns very high-level, abstract features describing complex relationships between words, such as grammar and cohesion. Additionally, we analyze linguistic-based features such as readability scores and the length of papers.

#### 4.4.1. Word-Level Insights from LR and RF

The coefficients learned by LR (see Figure 5a) and feature importance learned by RF indicate that *real* papers draw from a diverse set of words and—more often than *fake* papers—make references to specific sections (`"section"`), other papers (`"et"` and `"al"`), or recent trends (`"recently"`). In contrast, *fake* papers tend to rely on one-size-fits-all vocabulary such as `"method"`, `"approach"`, or `"implications"` more than *real* papers.

#### 4.4.2. LIME and SHAP Insights for Galactica, RoBERTa, and GPT-3

We use LIME [65] and SHAP [66] to inspect predictions made by Galactica, RoBERTa, and GPT-3. While these explanations fail to convey a concise overview, they are still useful to notice patterns and similarities across samples sharing labels and sources [67,68].

Often, RoBERTa and Galactica models tend to classify papers as *real* when the papers include infrequent words and sentences starting with adverbs. In addition, we notice that SHAP explanations corresponding to *real* papers have all words with low Shapley values. We believe this is intuitive as a paper appears *real* if it does not contain any artifact that strongly signals an AI source.

On the other hand, papers whose sections begin with *"In this paper, . . . "*, *"In this work, . . . "*, or *"In this study, . . . "* are often marked as *fake*. The same goes for those containing repeated words, spelling mistakes, or word fragments such as *"den"*, *"oly"*, *"um"*. Detectors are also able to spot incoherent content and context, as well as sections that are unnaturally short and do not convey any specific point. Several explanation instances of Galactica and RoBERTa can be found in Appendix C for further inspection. We choose not to provide an explanation for our GPT-3 classifier since it requires many requests to OpenAI's paid API.

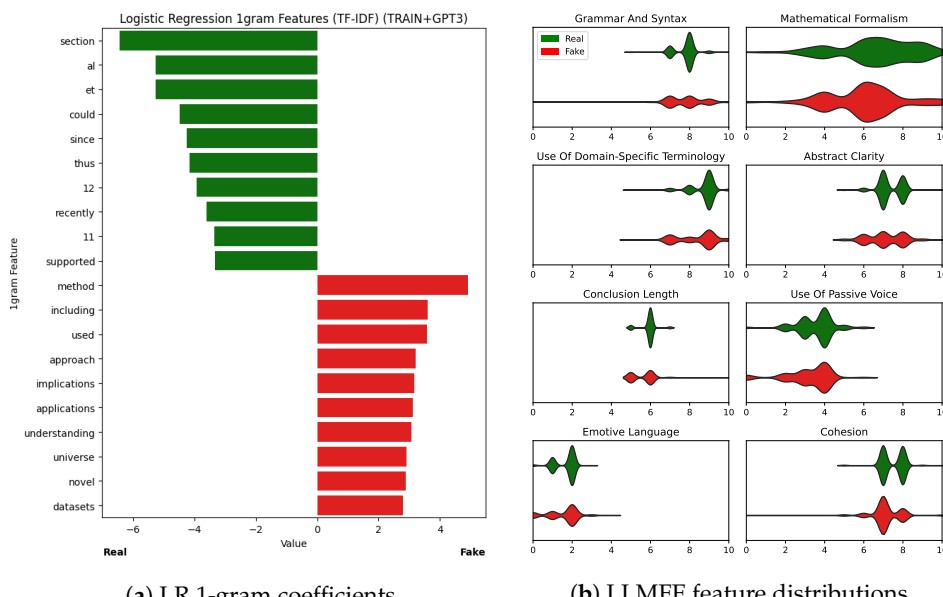

(**a**) LR 1-gram coefficients

(**b**) LLMFE feature distributions

**Figure 5.** Explainability insights from our Logistic Regression (LR) and Large Language Model Feature Extractor (LLMFE) classifiers. (**a**) shows the 1-grams with the 10 lowest (indicating *real*) and highest (indicating *fake*) coefficients learned by LR. (**b**) shows the distributions of scores for the eight most important features (according to Random Forest feature importance) learned by LLMFE.

### 4.4.3. Abstract Features from LLMFE

LLMFE identifies more abstract features such as *grammar and syntax*, *use of domain-specific terminology*, or *cohesion* as shown in Figure 5b. We observe that score distributions of *real* papers tend to be narrower than those of *fake* papers. This is not surprising given that *fake* papers were generated by multiple generators, some more and some less advanced. For many features, the distributions of *real* and *fake* papers have the same mode, suggesting that collectively our dataset of machine-generated papers resembles *real* papers quite well.

### 4.4.4. Readability Metrics for Different Generators

Figure 6 shows the distribution of Flesch–Kincaid Grade Level [69] and Gunning Fog [70] readability metrics [71] for papers from the different generators and *real* papers. Flesch–Kincaid measures the technical difficulty of the papers, while Gunning Fog measures the readability of the papers. The comparison confirms our observation that our machine-generated papers are representative of *real* papers with a slight increase in writing sophistication from SCIgen and GPT-2 to ChatGPT and GPT-3 generators, with Galactica being the median.

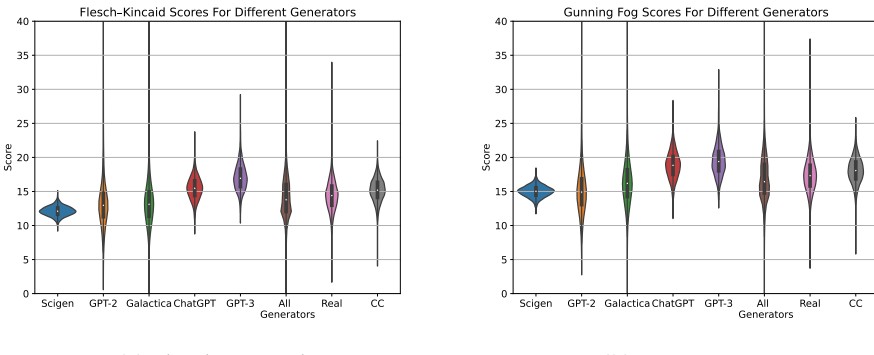

(**a**) Flesch Kincaid

(**b**) Gunning Fog

**Figure 6.** Distribution of readability metrics for papers from the different generators. (**a**) shows Flesch–Kincaid scores while (**b**) shows Gunning Fog scores for all generators.

### 4.4.5. Generated Texts Length

We observe differences in the length of the sections in our *fake* scientific papers depending on the generator. Figure 7 shows the length distributions of sections generated by the different generators. On average, machine-generated sections from all generators are shorter than sections from *real* papers—the only exception being abstracts and conclusions generated by GPT-2, which are slightly longer than *real* abstracts and conclusions, on average. For most generators, we also see less length variety compared to real papers.

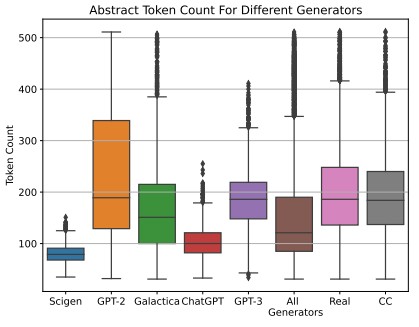

(**a**) Abstract Length

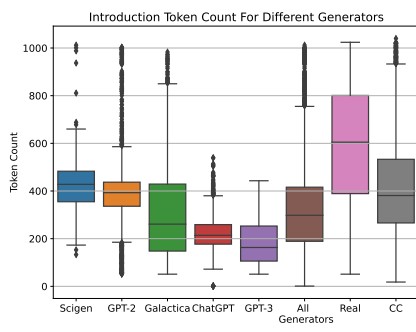

(**b**) Introduction Length

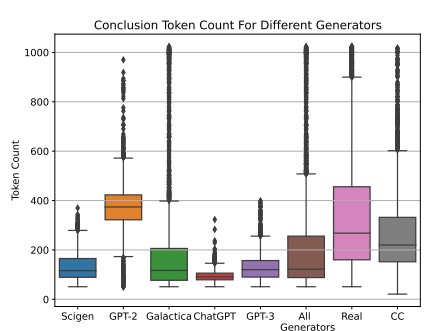

(**c**) Conclusion Length

**Figure 7.** The generators exhibit different tendencies for the length of the generated *fake* scientific papers. (**a**) shows the length distribution of generated abstracts, (**b**) shows the same for introductions, and (**c**) shows conclusion lengths.

For the co-created scientific papers (CC), despite prompting ChatGPT to return paraphrased sections with a similar length or even the exact word count as the original sections, we observe a tendency of ChatGPT to summarize sections during paraphrasing. While paraphrased abstracts have roughly the same length as their originals, paraphrased introductions, and conclusions sections are often significantly shorter, as shown in Figure 8. We conclude that ChatGPT does not reliably follow length constraints when confronted with a *paraphrasing* task on longer texts.

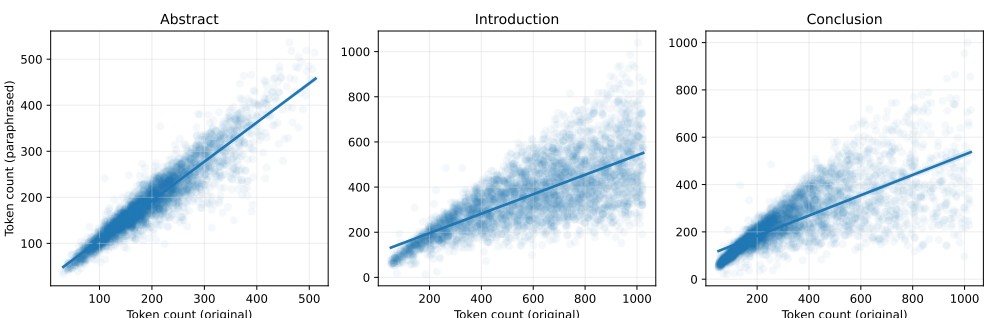

**Figure 8.** Paraphrasing sections with ChatGPT has a tendency to result in sections shorter than the original. The reduction in section length is most visible for the longer introduction and conclusion sections. For an analysis of lengths of generated *fake* scientific papers, see Figure 7 in the appendix.

## 5. Limitations and Future Work

Despite memory, GPU, and API limitations presenting significant obstacles for our project, we could still create high-quality *fake* scientific papers. Nonetheless, we believe there is room for improvement in addressing such limitations. For instance, beyond simply improving the quality of the generated papers, further insights could be gained from exploring generation processes entailing a collaboration between different models and input prompts.

Due to the complexity of parsing PDFs, we are currently limited to specific sections (abstract, introduction, conclusion) instead of complete papers. Moreover, processing entire publications would require substantial computational efforts. We believe that selecting sections dynamically at random instead of a fixed choice is worth exploring and will be the focus of future work.

Beyond DetectGPT [18], other zero-shot text detectors such as GPTZero (https://gptzero.me) (accessed on 31 July 2023) present promising solutions worth testing on our benchmark dataset. However, at the time of writing, such solutions are not available for experiments at scale.

In future work, we aim to address these limitations by exploring dynamic section selection, combining models and prompts in the generation process, improving papers' quality, and investigating the potential of zero-shot text detectors such as GPTZero as they become more accessible and scalable. We think that future research should further investigate how stable classifiers, such as the ones presented in this paper, are against newly appearing LLMs and how to improve the classifiers' generalization capabilities to out-of-domain samples.

## 6. Discussion, Ethical Considerations, and Broader Impact

It is important to emphasize that our work does not condemn the usage of LLMs. The legitimacy of their usage should be addressed by regulatory frameworks and guidelines. Still, we strongly believe it is crucial to develop countermeasures and strategies to detect machine-generated papers to ensure accountability and reliability in published research.

Our benchmark dataset serves as a valuable resource for evaluating detection algorithms, contributing to the integrity of the scientific community. However, potential challenges include adversarial attacks and dataset biases [72,73]. It is essential to develop robust countermeasures and strive for a diverse, representative dataset.

## 7. Conclusions

This work introduced a benchmark dataset for identifying machine-generated scientific papers in the LLM era. Our work creates a resource that allows researchers to evaluate the effectiveness of detection methods and thus support the trust and integrity of the scientific process.

We generated a diverse set of papers using both SCIgen and state-of-the-art LLMs—ChatGPT, Galactica, GPT-2, and GPT-3. This ensures a variety of sources and includes models capable of generating convincing content. We fine-tuned and tested several baseline detection models—Logistic Regression, Random Forest, GPT-3, Galactica, and RoBERTa—and compared their performance to DetectGPT, ChatGPT, and a novel Large Language Model Feature Extractor (LLMFE) that we propose. The results demonstrated varying degrees of success, with some models showing remarkable performance on specific subsets while sometimes struggling with out-of-domain data.

By providing a comprehensive platform for evaluating detection techniques, we contribute to the development of robust and reliable methods for identifying machine-generated content. Moving forward, we plan to address the current limitations and further enhance the utility of our benchmark for the research community.

We release a repository containing our benchmark dataset as well as the code used for experimental results (https://huggingface.co/datasets/tum-nlp/IDMGSP) (accessed on 31 July 2023).

**Author Contributions:** Design of experiments, E.M. and D.D.; Dataset creation, E.M., M.H.I.A. and S.M.; Experiments, M.H.I.A. and S.M.; writing—original draft preparation, E.M.; writing—journal version preparation, D.D. and S.M.; writing—review and editing, G.G. All authors have read and agreed to the published version of the manuscript.

**Funding:** This paper has been supported by the German *Federal Ministry of Education and Research* (BMBF, grant 01IS17049).

**Institutional Review Board Statement:** Not applicable.

**Data Availability Statement:** All data created within this research is available online (https://huggingface.co/datasets/tum-nlp/IDMGSP, accessed on 27 July 2023).

**Acknowledgments:** We would like to acknowledge the help of Social Research Computing Group for providing an opportunity to conduct this research and Paolo Basso and Margherita Musumeci for participating in the first version of this publication. Additionally, we would like to thank Leslie McIntosh for her guidance throughout our research journey.

**Conflicts of Interest:** The authors declare no conflict of interest.

## Abbreviations

The following abbreviations are used in this manuscript:

| | |
|---|---|
| NLP | Natural Language Processing |
| LLM | Large Language Model |
| BoW | Bag-of-Words |
| OOD | Out-of-Domain |
| CFG | Context-Free-Grammars |

## Appendix A. Generation Examples

In this section, we present examples of text that were generated using the models we employed.

### *Appendix A.1. ChatGPT*

Figures A1 and A2 provide examples of prompts that can be used to generate paper sections using ChatGPT.

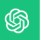

write a document with the title 'Video (language) modeling: a baseline for generative models of natural videos' with an abstract, an introduction and a conclusion.

Title: Video (Language) Modeling: A Baseline for Generative Models of Natural Videos

Abstract:
Generative models of natural videos have gained attention in recent years, as they enable a range of applications such as video synthesis, compression, and prediction. However, designing such models is a challenging task due to the high dimensionality and complexity of videos. In this paper, we propose a baseline approach for generative models of natural videos by modeling them as language sequences. Specifically, we encode video frames as vectors and train a language model on the resulting sequence of vectors. Our experimental results demonstrate that this approach is effective in generating realistic video frames, as well as in predicting future frames. We also compare our approach with other state-of-th art generative models and show its advantages and limitations.

Introduction:
Generative models of natural videos are becoming increasingly important in the field of computer vision, as they enable a wide range of applications such as video synthesis, compression, and prediction. However, designing such models is a challenging task due to the high dimensionality and complexity of videos. Traditional video models rely on hand-crafted features or require expensive computations. Recent advances in deep learning have led to the development of generative models that learn to generate videos from scratch, such as Generative Adversarial Networks (GANs) and Variational Autoencoders (VAEs). However, these models are still limited in their ability to generate high-quality videos that are both realistic and diverse.

In this paper, we propose a new baseline approach for generative models of natural videos. Our approach is based on modeling videos as language sequences, which allows us to leverage the advances in natural language processing (NLP) and apply them to the video domain. Specifically, we encode video frames as vectors and train a language model on the resulting sequence of vectors. Our experimental results demonstrate that this approach is effective in generating realistic video frames, as well as in predicting future frames. Moreover, our approach is simple, computationally efficient, and can be easily extended to incorporate other modalities such as audio.

Conclusion:
In this paper, we have proposed a new baseline approach for generative models of natural videos based on modeling videos as language sequences. Our approach is simple, computationally efficient, and achieves state-of-the-art results in generating realistic video frames and predicting future frames. We have also compared our approach with other state-of-the-art generative models and shown its advantages and limitations. Future work may include incorporating additional modalities, such as audio, and exploring new architectures that can further improve the quality and diversity of generated videos. Our approach opens up new possibilities for generative models of natural videos, and we hope that it will inspire further research in this direction.

**Figure A1.** Example 1 of prompting ChatGPT to produce sections of a scientific paper given the paper title.

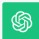

MO  write a document with the title 'A new characterization of the jump rate for piecewise-deterministic Markov processes with discrete transitions' with an abstract, an introduction and a conclusion.

Title: A new characterization of the jump rate for piecewise-deterministic Markov processes with discrete transitions

Abstract:
In this paper, we present a new characterization of the jump rate for piecewise-deterministic Markov processes with discrete transitions. We derive an explicit formula for the jump rate using a novel approach based on the time-reversibility of the process. Our formula is computationally efficient and can be used to analyze a wide range of systems, including biological and ecological models.

Introduction:
Piecewise-deterministic Markov processes (PDMPs) have been widely used to model complex systems in biology, physics, and engineering. PDMPs are characterized by a continuous-time deterministic evolution interspersed with random jumps. The jumps are determined by a Markov process with a transition rate that depends on the deterministic state of the system. In many applications, the transition rate is not known explicitly, and its estimation is a challenging problem.

In this paper, we propose a new approach to characterize the jump rate for PDMPs with discrete transitions. Our approach is based on the time-reversibility of the process, which allows us to derive an explicit formula for the jump rate. This formula is computationally efficient and does not require numerical simulations, making it an attractive tool for analyzing PDMPs in various fields.

Conclusion:
We have presented a new characterization of the jump rate for PDMPs with discrete transitions. Our approach is based on the time-reversibility of the process, which allows us to derive an explicit formula for the jump rate. Our formula is computationally efficient and can be used to analyze a wide range of systems, including biological and ecological models. believe that our results will be useful for researchers working with PDMPs, and we hope that our approach will stimulate further research in this field.

**Figure A2.** Example 2 of prompting ChatGPT to produce sections of a scientific paper given the paper title.

*Appendix A.2. Galactica*

For generating text with the Galactica model, an overview of the hyperparameters used is provided in Table A1. Each subsection shows how Galactica is prompted to generate the abstract, the introduction and the conclusion sections.

**Table A1.** Hyperparameters used to generate each paper section in the Galactica model. Each row corresponds to a decoding of a section based on the previous input sections. Here we used parameters of the MODEL.GENERATE function provided by Huggingface [74].

| Input Section(s) | Output Section | Tokenizer Max_Input_Size | Max_New_Tokens | Do_Sample | Temperature | top_k | top_p | No_Repeat_Ngram_Size | Early_Stopping |
|---|---|---|---|---|---|---|---|---|---|
| Title | Abstract | 64 | 512 | True | - | - | - | - | - |
| Title + Abstract | Introduction | 576 (64 + 512) | 1024 | True | 0.7 | 25 | 0.9 | 10 | True |
| Title + Abstract + Introduction | Conclusion | 1600 (64 + 512 + 1024) | 1024 | True | 0.7 | 25 | 0.9 | 10 | True |

Appendix A.2.1. Abstract Generation

**INPUT:**

Title: On the Global Structure of Hopf Hypersurfaces in Complex Space Form

**OUTPUT:**

Abstract: For a generic hypersurface in complex space form, all Hopf hypersurfaces are proved to be homogeneous or Hopf homogeneous. As a consequence, it is shown that the generalized Sasakian metric is a Ricci soliton on a Hopf hypersurface.

Appendix A.2.2. Introduction Generation

**INPUT:**

Title: On the Global Structure of Hopf Hypersurfaces in Complex Space Form

Abstract: For a generic hypersurface in complex space form, all Hopf hypersurfaces are proved to be homogeneous or Hopf homogeneous. As a consequence, it is shown that the generalized Sasakian metric is a Ricci soliton on a Hopf hypersurface.

Introduction:

**OUTPUT:**

A Hopf hypersurface is a hypersurface which admits a Hopf field, that is, a vector field such that $[\xi, \xi]$ is a normal vector field. The Hopf field is unique up to multiplication by a positive constant. The existence of Hopf fields is equivalent to the existence of a 1-form $\eta$ such that $\eta \wedge d\eta^{n-1}$ is a volume form. Let $M^n$ be a hypersurface in a complex space form $M^{2n+1}(c)$ with constant sectional curvature $c$. We denote by $g$ the metric induced by the Riemannian metric of $M^{2n+2}(c)$. The Hopf field is given by $\xi = \frac{1}{\sqrt{n}} J\nabla f$, where $J$ is the complex structure on $M^{2n+3}$ and $f$ is a function on $M^n \subset M^{2n+1}$. In [1], Wang and Chen showed that a Hopf hypersurface in $M^{2n+l}(c)$ is homogeneous if and only if the Hopf field is parallel. In this paper, we study the structure of Hopf hypersurfaces in complex space form. We show that a Hopf hypersurface is homogeneous if and only if it is Hopf homogeneous. We also show that a Hopf hypersurface is Hopf homogeneous if and only if the generalized Sasakian metric is a Ricci solitons on it.

Appendix A.2.3. Conclusion Generation

**INPUT:**

Title: On the Global Structure of Hopf Hypersurfaces in Complex Space Form

Abstract: For a generic hypersurface in complex space form, all Hopf hypersurfaces are proved to be homogeneous or Hopf homogeneous. As a consequence, it is shown that the generalized Sasakian metric is a Ricci soliton on a Hopf hypersurface.

Introduction:

A Hopf hypersurface is a hypersurface which admits a Hopf field, that is, a vector field such that $[\xi, \xi]$ is a normal vector field. The Hopf field is unique up to multiplication by a positive constant. The existence of Hopf fields is equivalent to the existence of a 1-form $\eta$ such that $\eta \wedge d\eta^{n-1}$ is a volume form. Let $M^n$ be a hypersurface in a complex space form $M^{2n+1}(c)$ with constant sectional curvature $c$. We denote by $g$ the metric induced by the Riemannian metric of $M^{2n+2}(c)$. The Hopf field is given by $\xi = \frac{1}{\sqrt{n}} J\nabla f$, where $J$ is the complex structure on $M^{2n+3}$ and $f$ is a function on $M^n \subset M^{2n+1}$. In [1], Wang and Chen showed that a Hopf hypersurface in $M^{2n+l}(c)$ is homogeneous if and only if the Hopf field is parallel. In this paper, we study the structure of Hopf hypersurfaces in complex space form. We show that a Hopf hypersurface is homogeneous if and only if it is Hopf homogeneous. We also show that a Hopf

hypersurface is Hopf homogeneous if and only if the generalized Sasakian metric is a Ricci solitons on it.

Conclusion:

**OUTPUT:**

We obtain a complete classification of Hopf hypersurfaces in complex space forms.

*Appendix A.3. GPT-2*

In the following, we show an example of how the fine-tuned GPT-2 was used to generate the abstract, the introduction, and the conclusion sections.

Appendix A.3.1. Abstract Generation

**INPUT:**

Competitive Multi-Agent Load Balancing with Adaptive Policies in Wireless Networks <|sep|>

**OUTPUT:**

In this paper, we propose adaptive multi-agent load balancing (AML) based on adaptive policies to achieve better performance. The adaptive policies can be designed to maximize the number of agent users and to reduce the number of latency. We present a comprehensive evaluation of AML on three wireless networks. We demonstrate that the adaptive policies outperform state-of-the-art AML strategies by an average of 6.5% and by an average of 2.5% when compared to state-of-the-art strategies. The performance of adaptive policies is evaluated on the three wireless networks with multiple users and multiple latency.

Appendix A.3.2. Introduction Generation

**INPUT:**

Competitive Multi-Agent Load Balancing with Adaptive Policies in Wireless Networks <|sep|>

**OUTPUT:**

Wireless networks are increasingly becoming the backbone of the modern world. The wireless networks have been the main source of energy for many years, and they are the main source of power for many businesses. However, wireless networks are also the main source of information for many people. For example, people in many countries rely on a variety of information services, such as social networks, social media, and e-mail. However, the information services are not available in all countries. Therefore, the information services are not available for all countries.

Appendix A.3.3. Conclusion Generation

**INPUT:**

Competitive Multi-Agent Load Balancing with Adaptive Policies in Wireless Networks <|sep|>

**OUTPUT:**

In this paper, we propose adaptive policies that optimize the network performance of different types of wireless networks. We show that adaptive policies can achieve competitive performance for different types of wireless networks. The proposed adaptive policies have been tested in several real-world wireless networks. We find that adaptive policies outperform state-of-the-art policies for the most common wireless networks. We also show that adaptive policies can be used to optimize the network performance of wireless networks by adapting their performance to different types of wireless networks. The proposed adaptive policies are able to achieve competitive performance for different types of wireless networks.

*Appendix A.4. GPT-3*

The following frames show how each section is generated using a GPT-3 generator.

Appendix A.4.1. Abstract Generation

**INPUT:**

Primitive Representation Learning for Scene Text Recognition

**OUTPUT:**

Primitive scene representation learning is an important subfield of deep learning, which has been widely applied in many image recognition tasks such as image segmentation and object detection. In recent years, several methods have been proposed to learn the primitive scene representation. Among them, most methods are based on U-Net architecture and use either a global or local context to capture the global or local structure of the input scenes. However, these approaches do not fully exploit the structure information in scenes. In this paper, we propose a novel primitive scene representation learning method for scene text recognition using a residual block-based encoder-decoder strategy with multiple level-up passes. To this end, we design three residual block structures with different subsampling ratios: 1:1 (global), 2:1 (local) and 1:2 (global+local). We also designed an attention module that can be used to further guide the search space in each level-up pass by enforcing certain conditions on hierarchical representations obtained during previous levels.

Appendix A.4.2. Introduction Generation

**INPUT:**

Primitive Representation Learning for Scene Text Recognition

Primitive scene representation learning is an important subfield of deep learning, which has been widely applied in many image recognition tasks such as image segmentation and object detection. In recent years, several methods have been proposed to learn the primitive scene representation. Among them, most methods are based on U-Net architecture and use either a global or local context to capture the global or local structure of the input scenes. However, these approaches do not fully exploit the structure information in scenes. In this paper, we propose a novel primitive scene representation learning method for scene text recognition using a residual block-based encoder-decoder strategy with multiple level-up passes. To this end, we design three residual block structures with different subsampling ratios: 1:1 (global), 2:1 (local) and 1:2 (global+local). We also designed an attention module that can be used to further guide the search space in each level-up pass by enforcing certain conditions on hierarchical representations obtained during previous levels.

Introduction:

**OUTPUT:**

[1,2]. It has been widely applied in many image segmentation and object detection [3,4] tasks. In these applications, deep learning is often used to learn the complex parameters such as models and training schemes. Alternatively, it can be also applied on the more fundamental problems of learning a better model for predicting new input data that have not been seen before. Learning representations of unseen data is one of the most fundamental problems within machine learning and has attracted tremendous attention in recent years [5]. Primitive scene representation learning (PSRL) is an important subfield of deep learning which aims at modeling the basic visual concepts such as objects and scenes in order to solve various real-world applications likeimage segmentation [6], pedestrian detection [7], video classification [8] etc.

Appendix A.4.3. Conclusion Generation

**INPUT:**

Primitive Representation Learning for Scene Text Recognition

Primitive scene representation learning is an important subfield of deep learning, which has been widely applied in many image recognition tasks such as image segmentation and object detection. In recent years, several methods have been proposed to learn the primitive scene representation. Among them, most methods are based on U-Net architecture and use either a global or local context to capture the global or local structure of the input scenes. However, these approaches do not fully exploit the structure information in scenes. In this paper, we propose a novel primitive scene representation learning method for scene text recognition using a residual block-based encoder-decoder strategy with multiple level-up passes. To this end, we design three residual block structures with different subsampling ratios: 1:1 (global), 2:1 (local) and 1:2 (global+local). We also designed an attention module that can be used to further guide the search space in each level-up pass by enforcing certain conditions on hierarchical representations obtained during previous levels.

[1,2]. It has been widely applied in many image segmentation and object detection [3,4] tasks. In these applications, deep learning is often used to learn the complex parameters such as models and training schemes. Alternatively, it can be also applied on the more fundamental problems of learning a better model for predicting new input data that have not been seen before. Learning representations of unseen data is one of the most fundamental problems within machine learning and has attracted tremendous attention in recent years [5]. Primitive scene representation learning (PSRL) is an important subfield of deep learning which aims at modeling the basic visual concepts such as objects and scenes in order to solve various real-world applications likeimage segmentation [6], pedestrian detection [7], video classification [8] etc.

**OUTPUT:**

We propose a novel residual block-based encoder-decoder (R-BiDe) method for learning the primitive scene representation. We introduce a residual block structure with different subsampling ratios, and design an attention module that can be used to guide the search space in each level-up pass. The proposed R-BiDe method achieves better performance than other stateof-theart methods on several difficult scene text recognition benchmarks with no overfitting issues. In future works, we will focus on exploring the model design principles of R-BiDe as well as its application scenarios such as object detection and autonomous driving using deep learning frameworks such as Caffe and PyTorch.

*Appendix A.5. SCIgen*

In the case of SCIgen papers, a script is used to generate the paper documents. Once the documents are generated, they are subsequently parsed to extract the relevant sections. The following frame shows an example of a generation after parsing.

Title: Towards the Understanding of Randomized Algorithms

Abstract: Many security experts would agree that, had it not been for online algorithms, the investigation of XML might never have occurred. Given the current status of heterogeneous epistemologies, statisticians predictably desire the evaluation of public-private key pairs, which embodies the important principles of programming languages. In order to solve this riddle, we present an analysis of expert systems (Judas), disconfirming that symmetric encryption and hash tables are never incompatible. It at first glance seems counterintuitive but fell in line with our expectations .

Introduction: Recent advances in interposable modalities and trainable modalities do not necessarily obviate the need for 802.11 mesh networks. After years of robust research into Byzantine fault tolerance, we disprove the evaluation of information retrieval systems. The notion that security experts collaborate with atomic symmetries is generally adamantly opposed. The evaluation of link-level acknowledgements would tremendously amplify vacuum tubes. The basic tenet of this solution is the development of IPv7. For example, many frameworks allow the study of the transistor. On a similar note, we emphasize that our system improves systems. Thus, we use lossless communication to disprove that online algorithms and journaling file systems can interact to fulfill this ambition. Relational methodologies are particularly unfortunate when it comes to "smart" information. This is an important point to understand. However, for example, many frameworks observe the memory bus. Thusly, we see no reason not to use trainable communication to develop concurrent theory. This outcome is usually a private ambition but is supported by related work in the field. We describe new perfect modalities, which we call Judas. Though such a claim might seem counterintuitive, it has ample historical precedence. To put this in perspective, consider the fact that little-known futurists never use IPv6 to surmount this problem. Contrarily, stochastic technology might not be the panacea that cyberneticists expected. Two properties make this method distinct: our application prevents homogeneous configurations, and also Judas is copied from the analysis of DHTs [1]. To put this in perspective, consider the fact that little-known information theorists rarely use 802.11b to address this challenge. Combined with the UNIVAC computer, such a claim synthesizes new stochastic modalities. The rest of this paper is organized as follows. Primarily, we motivate the need for the transistor. Similarly, we place our work in context with the prior work in this area. On a similar note, to solve this question, we construct an analysis of telephony (Judas), which we use to show that the seminal relational algorithm for the exploration of active networks by Thompson [1] runs in $\Omega(\log \log n)$ time. In the end, we conclude.

Conclusion: Our method will address many of the issues faced by today's theorists. Similarly, Judas can successfully prevent many link-level acknowledgements at once. Our methodology for constructing the improvement of the Turing machine is particularly excellent. We plan to explore more problems related to these issues in future work.

## Appendix B. Classifier Details

*Appendix B.1. Bag-of-Words Classifiers*

Table A2 shows the detailed results for the different bag-of-words classifiers introduced in Section 4.2.1.

**Table A2.** Experiment results for the different bag-of-words classifiers reported with accuracy metric. Out-of-domain experiments are highlighted in blue. The highest values per test set are highlighted in bold.

| Model | Train Dataset | TEST | OOD-GPT3 | OOD-REAL | TECG | TEST-CC |
|-------|--------------|------|----------|----------|------|---------|
| LR-1gram (tf-idf) | TRAIN | 95.3% | 4.0% | 94.6% | 96.1% | 7.8% |
| LR-1gram (tf-idf) | TRAIN + GPT3 | 94.6% | 86.5% | 86.2% | 97.8% | 13.7% |
| LR-1gram (tf-idf) | TRAIN-CG | 86.6% | 0.8% | 97.8% | 32.6% | 1.2% |
| LR-2gram (tf-idf) | TRAIN | 89.1% | 0.5% | 96.5% | 91.3% | 6.4% |
| LR-2gram (tf-idf) | TRAIN + GPT3 | 90.0% | 89.7% | 86.1% | 97.3% | 15.7% |
| LR-2gram (tf-idf) | TRAIN-CG | 73.3% | 0.0% | **99.6%** | 1.4% | 0.6% |
| LR-(1,2)gram (tf-idf) | TRAIN | 94.8% | 0.2% | 97.8% | 94.6% | 2.7% |
| LR-(1,2)gram (tf-idf) | TRAIN + GPT3 | 95.1% | 83.3% | 92.6% | 97.8% | 5.9% |
| LR-(1,2)gram (tf-idf) | TRAIN-CG | 83.3% | 0.2% | 99.3% | 1.7% | 0.3% |
| RF-1gram (tf-idf) | TRAIN | 94.8% | 24.7% | 87.3% | **100.0%** | 8.1% |
| RF-1gram (tf-idf) | TRAIN + GPT3 | 91.7% | 95.0% | 69.3% | **100.0%** | 15.1% |
| RF-1gram (tf-idf) | TRAIN-CG | **97.6%** | 7.0% | 95.0% | 57.0% | 1.7% |
| RF-2gram (tf-idf) | TRAIN | 90.8% | 12.4% | 76.8% | 99.3% | 29.9% |
| RF-2gram (tf-idf) | TRAIN + GPT3 | 87.7% | **96.8%** | 54.6% | 99.9% | **44.0%** |
| RF-2gram (tf-idf) | TRAIN-CG | 85.8% | 3.4% | 88.8% | 44.1% | 8.5% |
| RF-(1,2)gram (tf-idf) | TRAIN | 95.4% | 22.4% | 87.8% | 93.8% | 9.1% |
| RF-(1,2)gram (tf-idf) | TRAIN + GPT3 | 93.8% | 96.0% | 66.6% | **100.0%** | 19.7% |
| RF-(1,2)gram (tf-idf) | TRAIN-CG | 87.8% | 1.9% | 96.8% | 43.8% | 1.1% |

*Appendix B.2. GPT-3*

The following frame shows a GPT-3 classifier training prompt. The input label (`1` for *fake* and `0` for *real*) is separated from the input by the separator token (`###`).

Abstract:

For a generic hypersurface in complex space form, all Hopf hypersurfaces are proved to be homogeneous or Hopf homogeneous. As a consequence, it is shown that the generalized Sasakian metric is a Ricci soliton on a Hopf hypersurface.

Introduction:

A Hopf hypersurface is a hypersurface which admits a Hopf field, that is, a vector field such that $[\xi, \xi]$ is a normal vector field. The Hopf field is unique up to multiplication by a positive constant. The existence of Hopf fields is equivalent to the existence of a 1-form $\eta$ such that $\eta \wedge d\eta^{n-1}$ is a volume form. Let $M^n$ be a hypersurface in a complex space form $M^{2n+1}(c)$ with constant sectional curvature $c$. We denote by $g$ the metric induced by the Riemannian metric of $M^{2n+2}(c)$. The Hopf field is given by $\xi = \frac{1}{\sqrt{n}} J \nabla f$, where $J$ is the complex structure on $M^{2n+3}$ and $f$ is a function on $M^n \subset M^{2n+1}$. In [1], Wang and Chen showed that a Hopf hypersurface in $M^{2n+l}(c)$ is

homogeneous if and only if the Hopf field is parallel. In this paper, we study the structure of Hopf hypersurfaces in complex space form. We show that a Hopf hypersurface is homogeneous if and only if it is Hopf homogeneous. We also show that a Hopf hypersurface is Hopf homogeneous if and only if the generalized Sasakian metric is a Ricci solitons on it.

Conclusion:

For a generic hypersurface in complex space form, all Hopf hypersurfaces are proved to be homogeneous or Hopf homogeneous. As a consequence, it is shown that the generalized Sasakian metric is a Ricci soliton on a Hopf hypersurface.

###

*Appendix B.3. ChatGPT*

Table A3 shows the detailed results for the different ChatGPT prompting styles introduced in Section 4.2.6.

**Table A3.** Experiment results for different ChatGPT prompting styles reported with accuracy metric. Out-of-domain experiments are highlighted in blue. Highest values per test set are highlighted in bold. (*) ChatGPT accuracies have been evaluated on randomly sampled subsets of 100 scientific papers per test set and prompting style due to API limits.

| Model | Train Dataset | TEST | OOD-GPT3 | OOD-REAL | TECG | TEST-CC |
|---|---|---|---|---|---|---|
| ChatGPT-IO [*] | - | **69%** | **49%** | 89% | 0% | 3% |
| ChatGPT-CoT [*] | - | 63% | 2% | 70% | 3% | 1% |
| ChatGPT-IP [*] | - | 57% | 18% | 92% | **7%** | **5%** |
| ChatGPT-FS [*] | TRAIN + GPT3 | 59% | 2% | **100%** | 0% | 0% |

*Appendix B.4. Large Language Model Feature Extractor (LLMFE)*

Figure A3 show an extract from the hierarchical clustering dendrogram learned during the feature consolidation step of LLMFE.

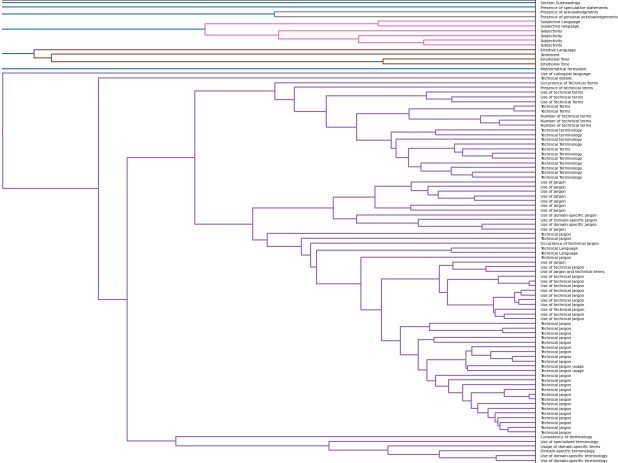

**Figure A3.** Extract from the hierarchical clustering dendrogram learned during the feature consolidation step of LLMFE. The full dendrogram lists all 884 features. The distance threshold was chosen so that 83 clusters were created from the 884 features.

## Appendix C. Explainability Results

*Appendix C.1. Bag-of-Words Classifiers*

Figures A4–A6 show the coefficients and feature importance learned by our Logistic Regression (LR) and Random Forest (RF) classifiers on the TRAIN, TRAIN + GPT3, and TRAIN-CG datasets, respectively.

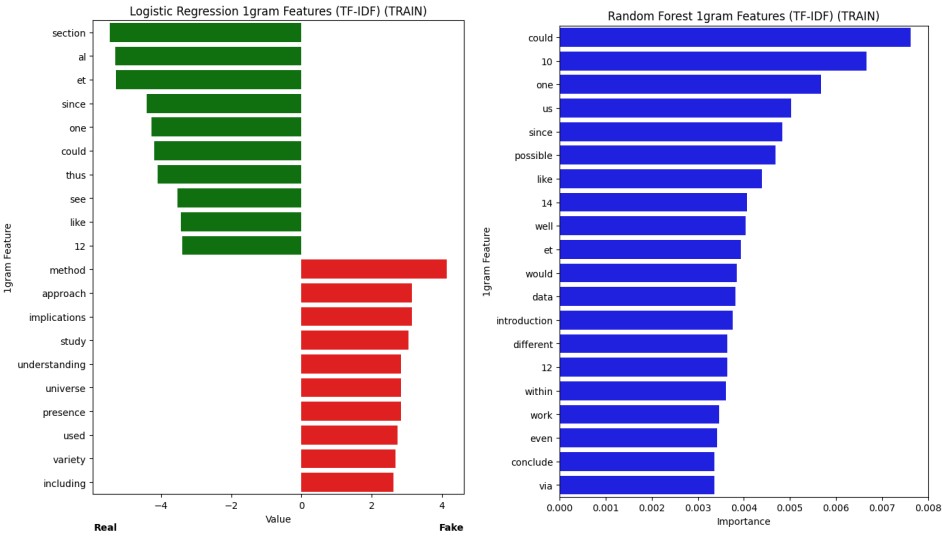

(**a**) LR 1-gram coefficients　　　　　　(**b**) RF 1-gram feature importance

**Figure A4.** Explainability insights from our Logistic Regression (LR) and Random Forest (RF) classifiers on the TRAIN dataset. (**a**) shows the 1-grams with the 10 lowest (indicating *real*) and highest (indicating *fake*) coefficients learned by LR. (**b**) shows the feature importance extracted from RF after training.

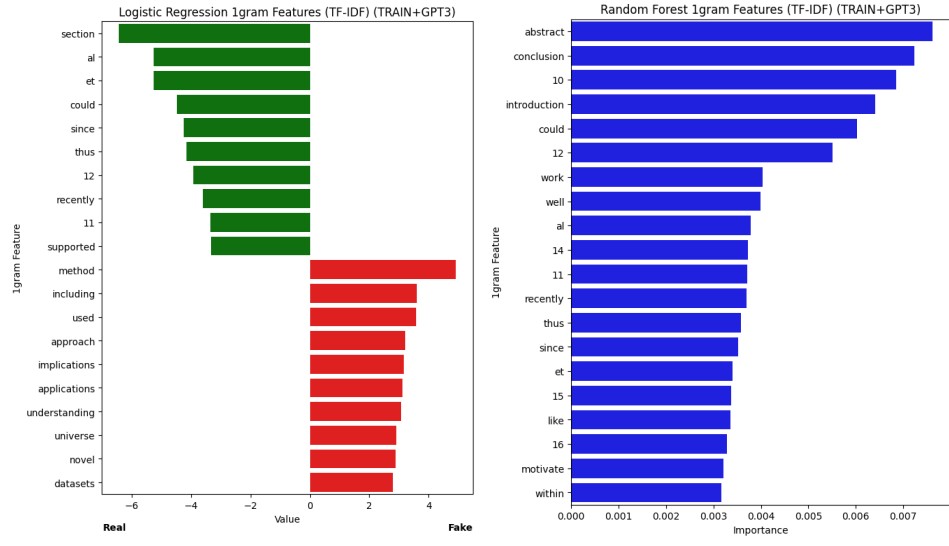

(**a**) LR 1-gram coefficients　　　　　　(**b**) RF 1-gram feature importance

**Figure A5.** Explainability insights from our Logistic Regression (LR) and Random Forest (RF) classifiers on the TRAIN + GPT3 dataset. (**a**) shows the 1-grams with the 10 lowest (indicating *real*) and highest (indicating *fake*) coefficients learned by LR. (**b**) shows the feature importance extracted from RF after training.

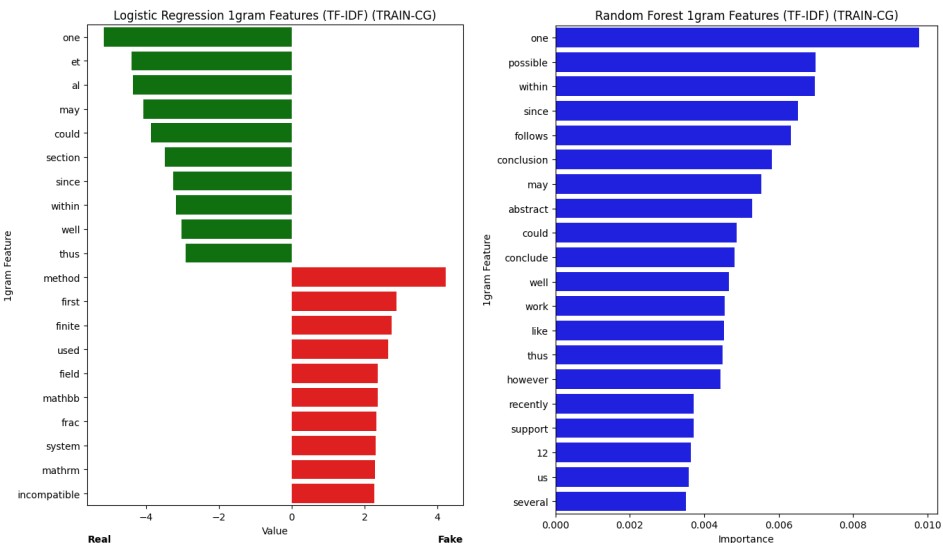

(**a**) LR 1-gram coefficients     (**b**) RF 1-gram feature importance

**Figure A6.** Explainability insights from our Logistic Regression (LR) and Random Forest (RF) classifiers on the TRAIN-CG dataset. (**a**) shows the 1-grams with the 10 lowest (indicating *real*) and highest (indicating *fake*) coefficients learned by LR. (**b**) shows the feature importance extracted from RF after training.

*Appendix C.2. RoBERTa*

Selected samples of SHAP and LIME explanations for our RoBERTa classifier can be found in Figures A7–A17.

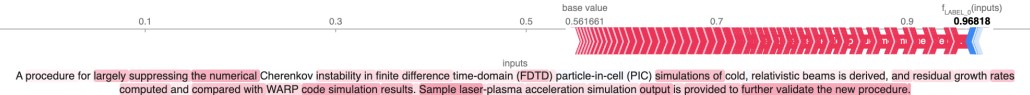

**Figure A7.** RoBERTa: Example of SHAP explanation on a real abstract correctly classified.

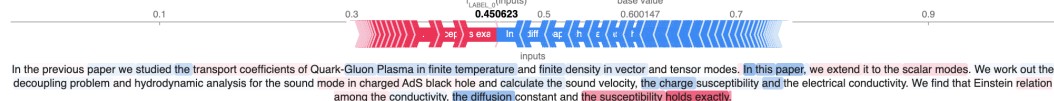

**Figure A8.** RoBERTa: Example of SHAP explanation on a real misclassified abstract.

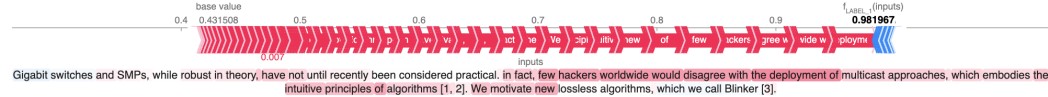

**Figure A9.** RoBERTa: Example of SHAP explanation on a SCIgen generated abstract correctly classified.

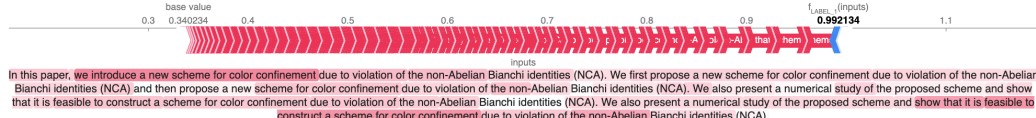

**Figure A10.** RoBERTa: Example of SHAP explanation on a GPT-2 generated abstract correctly classified.

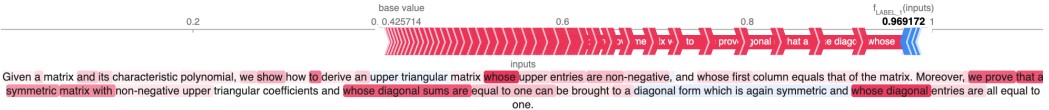

**Figure A11.** RoBERTa: Example of SHAP explanation on a Galactica generated abstract correctly classified.

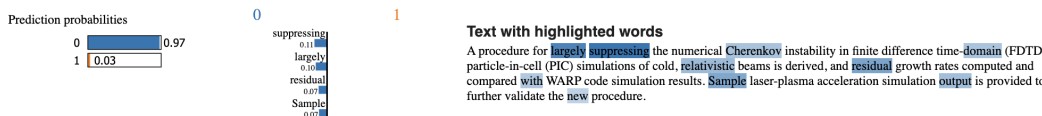

**Figure A12.** RoBERTa: Example of SHAP explanation on a ChatGPT generated abstract correctly classified.

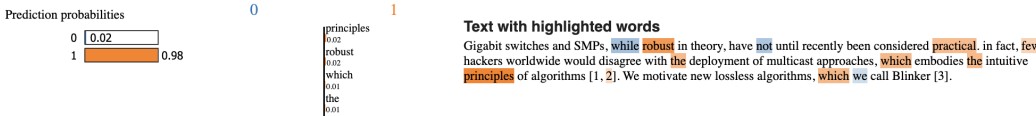

**Figure A13.** RoBERTa: Example of LIME explanation on a real abstract correctly classified.

**Figure A14.** RoBERTa: Example of LIME explanation on a SCIgen generated abstract correctly classified.

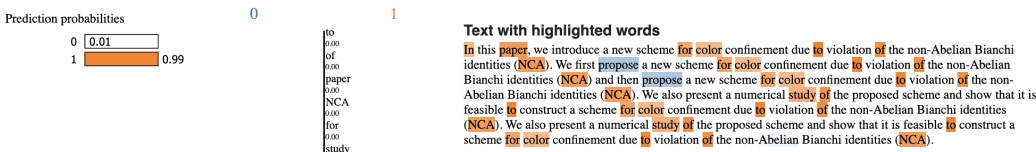

**Figure A15.** RoBERTa: Example of LIME explanation on a GPT-2 generated abstract correctly classified.

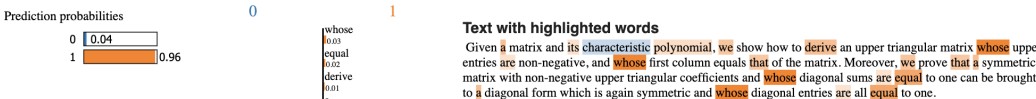

**Figure A16.** RoBERTa: Example of LIME explanation on a Galactica generated abstract correctly classified.

**Figure A17.** RoBERTa: Example of LIME explanation on a ChatGPT generated abstract correctly classified.

*Appendix C.3. Galactica*

Selected samples of SHAP explanations for our Galactica classifier can be found in Figures A18–A21.

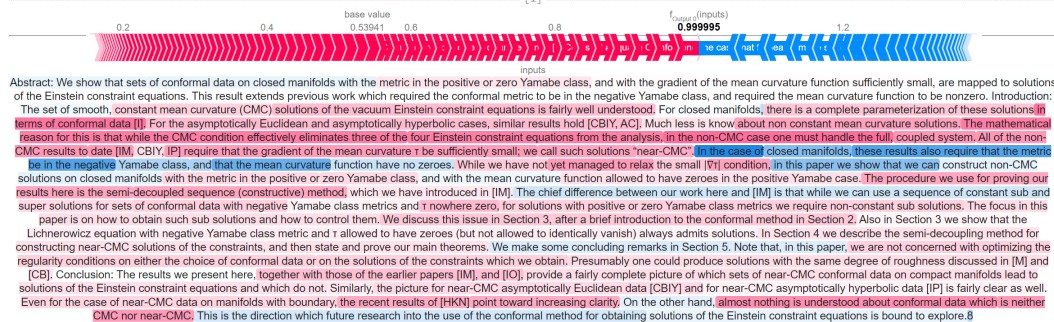

**Figure A18.** Galactica: Example of SHAP explanation on a real paper correctly classified.

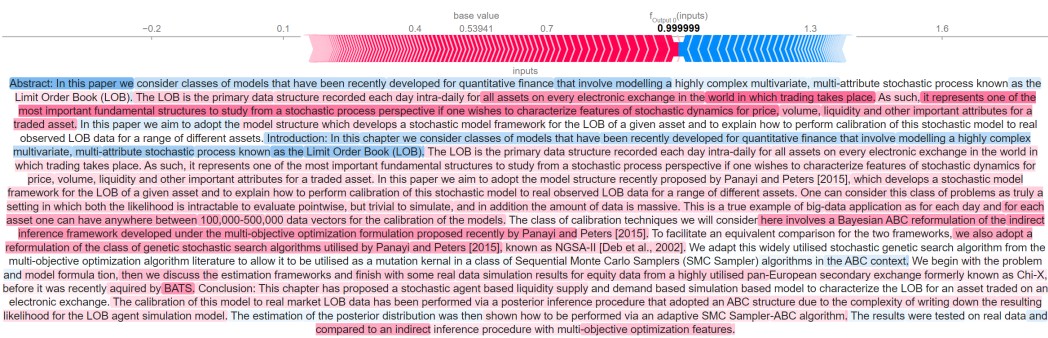

**Figure A19.** Galactica: Example of SHAP explanation on a misclassified real paper.

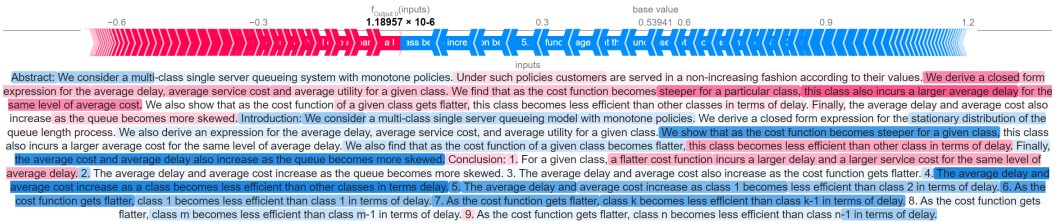

**Figure A20.** Galactica: Example of SHAP explanation on a Galactica generated paper correctly classified.

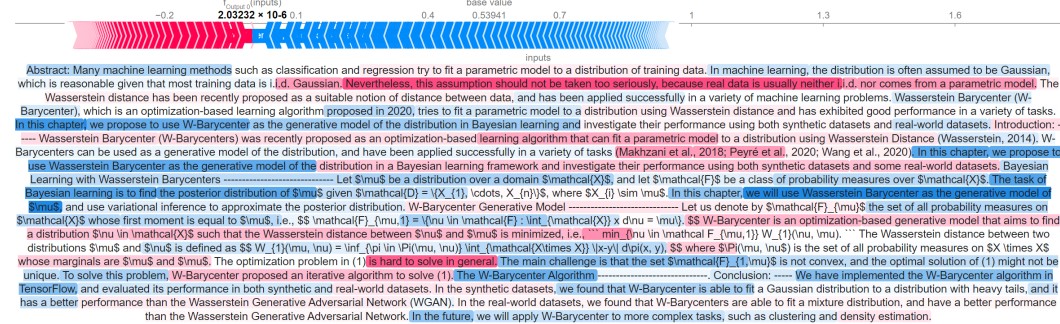

**Figure A21.** Galactica: Example of SHAP explanation on a misclassified Galactica generated paper.

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
