# Peer review of "A Benchmark Dataset to Distinguish Human-Written and Machine-Generated Scientific Papers"

_information, doi:10.3390/info14100522_

Round 1

Reviewer 1 Report

In recent years, generative natural language processing (NLP) systems based on large language models (LLMs) have made tremendous progress, with state-of-the-art algorithms capable of generating text that is nearly indistinguishable from human-written text. This advancement has been widely used in various fields, such as chatbots, automatic content generation, and summarization tools. However, it also raises doubts about the integrity and authenticity of academic writing and scientific publishing.

This paper mainly uses various Large Language Models (LLM) to distinguish papers written by real people, machines, and human-machine co-authors. Each paper only uses abstract, introduction, and conclusion for analysis, and finally analyzes The pros and cons of each LLM.

Main contributions:

1. A data set is proposed, including three types of documents written by real people, written by machines, and co-written by humans and machines. Among them, the documents written by real people are mainly collected from the paper data set of arXiv. For files written by machines, the titles of papers written by real people are input into SCIgen, ChatGPT, Galactia, GPT-2, and GPT-3, and the corresponding abstract, introduction, and conclusion are generated. The document co-written by man and machine is rewritten by ChatGPT using 4000 papers from arXiv.

2. Use seven different classifiers including: SCIgen, GPT-2, GPT-3, ChatGPT and Galactica and use LLM+RF and LLM+LR to classify files. In classifying scientific papers produced by GPT3, LR and RF have up to 90% accuracy. However, the accuracy of LR and RF in other fields of papers produced by GPT3 has dropped significantly. All models can perform well using the datasets of Train+GPT3 during training.

3. Emphasis on the generalization ability and interpretability of experiments to gain insight into the strengths and weaknesses of each detector and provide relevant insights. Papers written by real people are more linguistically diverse than other papers, citing specific sections ("section"), other papers ("et" and "al"), or recent trends ("recently") more often. Papers written by machines tend to use some general vocabulary, such as "method", "approach" or "implications".

Here are some general comments:

1. Since it is aimed at distinguishing between papers written by real people and machines, the authors should prove that the papers on arXiv are human written rather than co-created or machine-generated.

2. In Table3, the significance of using color emphasis is not mentioned in this article. Among them, the results of TEST-CC: human-machine co-created papers are quite unreasonable and should be explained.

none

Author Response

Dear Reviewer,

Thank you for pointing out important flaws in our methodology and results description. Here is the explanation of how we have addressed your questions:

  1. The snapshot of arxiv dataset from Kaggle were cleaned to take only the papers before November, 2022. We added this information to the manuscript, section 3.1. Of course, there could be still cases when scientists used some generative or paraphrasing tools before. However, we believe that before ChatGPT, the impact of these types of generation  was insignificant.
  2. We rephrased explanations in the caption of Table 3 and added some changes in section 4.3.

Reviewer 2 Report

To distinguish between machine-generated content and human-written text, this article introduces a novel benchmark dataset and experiments with multiple types of classifiers to highlight the advantages and disadvantages of different classifiers. But there are still some issues with this article:

1.The authors do not explain why language models such as SCIgen were chosen. It is suggested to explain in combination with relevant work to increase persuasiveness.

2.Innovation point 3 in the introduction does not indicate what problem is solved."Provide insights into the strengths and weaknesses of each detector" overlaps with innovation point 2.

3.It was pointed out in 3.3 that rewriting paragraphs using ChatGPT makes them shorter. The conclusion generated by Galactica in the appendix only has one sentence. Is it possible to add relevant experiments as shown in Figure 4 for different language models?

4.More complex algorithms, such as LLMFR in 4.2.7, suggest using formulas to explain the input and output of the corresponding features, and modify Figure 5 accordingly.

5.The explanation of the results in 4.3 is incomplete. The article explains why LLMFE outperformed ChatGPT on all datasets except OOD-REAL, but does not explain why Roberta performed better on test-CC and TRCG datasets.

6.In Figure 7, only Flesh-Kincaid scores and Gunning Fog scores of ood_real and gpt3 are higher than those of real articles. So, the increase in writing complexity cannot be seen.

7.Is an example of GPT-3 generation missing from the appendix?

8.Some recent articles are valued to refer, such as:

An Effective Learning Evaluation Method Based on Text Data with Real-time Attribution - A Case Study for Mathematical Class with Students of Junior Middle School in China, ACM Transactions on Asian and Low-Resource Language Information Processing, 2023, 22(3): 63

Author Response

Dear Reviewer,

Thank you for pointing out important flaws in our methodology and results description. Here is the explanation of how we have addressed your questions:

  1. We extended the explanation of Scigen usage in the beginning of section 3.2.1.
  2. We rephrased point 3 of the contributions to make it more specific.
  3. We added a new section 4.4.5 Generated Texts Length Analysis with the Figure 7 showing the differences in separate generated parts from used generators.
  4. We redraw Figure 4 with more clear steps separation of the proposed pipeline.
  5. We extended the results discussion in section 4.3. 
  6. We redraw Figure 6 changing the scale so the difference in metrics means across the models is more visible.
  7. We added all models prompting and generation examples to Appendix A.
  8. We include the proposed relevant publication to the Related Work section.

Reviewer 3 Report

I would like to thank both the editor and authors for granting me the opportunity to review this manuscript. The paper presents an innovative benchmark dataset aimed at distinguishing between human-generated and machine-generated contexts. After a thorough review, I recommend its acceptance pending minor revisions. Below are my comments:

1. The topic of identifying machine-generated contexts has gained substantial traction, as evidenced by recent works like the one presented in arXiv:2301.10226 at ICML 2023. While watermark techniques are often facilitated by model providers, this paper focus on identifying machine-generated contexts from the user's standpoint, a commendable approach.

2. The authors have invested significant effort in curating the benchmark dataset. Particularly noteworthy is the creation of co-create datasets, enhancing the complexity of the classification problem. This open-sourced dataset is a great contribution for future academic exploration.

3. The manuscript presents a comprehensive array of experiments involving classifiers, ranging from conventional machine learning approaches to cutting-edge Large Language Models (LLMs). The experimental design is well-considered and thought-provoking.

However, there exist room for refinement:

1. While the manuscript furnishes a suitable background and introduction, given the current surge in LLM research, it may benefit from citing pivotal papers in the field. For instance, in Line 15, apart from referencing the OpenAI paper, acknowledging the significance of Llama 2 (Meta, arXiv:2307.09288) and PaLM 2 (Google, arXiv:2305.10403) could enrich the contextualization.

2. Although the paper adeptly constructs synthetic datasets using diverse models, it appears the authors did not explore the prospect of aggregating various models to create synthetic datasets. For example, use GPT-2 to generate abstract, and then input the title and abstract to GPT-3 to generate introduction, then use Galactica to generate conclusion. This can diminish the individual bias of each model and make the dataset more complex to distinguish from human.

3. Enhancing dataset complexity could also be achieved by introducing diverse prompts for generating synthetic data. Some prompts (e.g. 'Write the introduction like a human researcher') may sometimes activate model's self-refine ability and make the synthetic datasets more challenging. I am wondering whether the authors explored this direction.

4. Figures 2. 'three separate models' -> I found it difficult to understand at the beginning. I suggest to be more explicit here, e.g. 'three separately fine-tuned GPT-2 models'

5. In Lines 249 to 253, the paper notes that 'paraphrased introduction and conclusion sections are often significantly shorter'. It would be insightful to explore techniques for mitigating this potential length bias. For instance, explicitly instructing LLMs to adhere to length constraints, such as 'Provide the conclusion in 400 words', could be an avenue to explore. I assume ChatGPT should be quite good at following the length constraints.

Author Response

Dear Reviewer,

Thank you for your interesting questions and ideas. We agree with you that some of your proposed experiments would be interesting to add to our work. However, due to limited time frame or the next submission, we incorporate some of your suggested edits and refer to other suggestions as future work:

  1. We added reference to the listed models in Introduction and Related Work sections.
  2. This is an extremely interesting experiment idea. However, such test set creation requires some extensive experiments. We mention such extension of our work in Section 5.
  3. and 5. During preliminary experiments, we experimented with several prompts. The choice of ones listed in the paper was based on the best generation results. We also tried to explicitly mention the desired length of the text. However, the generators did nor catch this requirement.  For instance, for ChatGPT, the official justification of this fact was published: https://community.openai.com/t/chatgpt-cannot-count-words-or-produce-word-count-limited-text/47380 We added the short explanation of prompt design in the manuscript. Also, we added a new section 4.4.5 Generated Texts Length Analysis with the Figure 7 showing the differences in separate generated parts from used generators.
  4. We rephrase the caption of Figure 2.

Round 2

Reviewer 2 Report

no other comments

Author Response

Dear Reviewer,

Thank you for all your comments and high evaluation of our work!